# Identifying Spatial Co-occurrence in Healthy and InflAmed tissues (ISCHIA)

Atefeh Lafzi[1,2,3], Costanza Borrelli [2,3], Simona Baghai Sain [2], Karsten Bach[2], Jonas A Kretz [2], Kristina Handler [2], Daniel Regan-Komito [1], Xenia Ficht [2], Andreas Frei [1] & Andreas Moor [2✉]

## Abstract

Sequencing-based spatial transcriptomics (ST) methods allow unbiased capturing of RNA molecules at barcoded spots, charting the distribution and localization of cell types and transcripts across a tissue. While the coarse resolution of these techniques is considered a disadvantage, we argue that the inherent proximity of transcriptomes captured on spots can be leveraged to reconstruct cellular networks. To this end, we developed ISCHIA (Identifying Spatial Co-occurrence in Healthy and InflAmed tissues), a computational framework to analyze the spatial co-occurrence of cell types and transcript species within spots. Co-occurrence analysis is complementary to differential gene expression, as it does not depend on the abundance of a given cell type or on the transcript expression levels, but rather on their spatial association in the tissue. We applied ISCHIA to analyze co-occurrence of cell types, ligands and receptors in a Visium dataset of human ulcerative colitis patients, and validated our findings at single-cell resolution on matched hybridization-based data. We uncover inflammation-induced cellular networks involving M cell and fibroblasts, as well as ligand-receptor interactions enriched in the inflamed human colon, and their associated gene signatures. Our results highlight the hypothesis-generating power and broad applicability of co-occurrence analysis on spatial transcriptomics data.

**Keywords** Spatial Transcriptomics; Co-occurrence Analysis; Cellular Networks; Ligand–Receptor Interaction; Ulcerative Colitis
**Subject Categories** Chromatin, Transcription & Genomics; Computational Biology; Methods & Resources

## Introduction

Tissue ecosystems are maintained by the co-existence and coordinated function of cellular networks (CNs). CNs are the basic functional unit of any tissue: communities of neighboring cells that interact to perform a physiological function, with cells as nodes and cell-cell interactions (CCIs) as edges. A prime example of CN is the colonic crypt, where a network of Wnt-secreting mesenchymal cells and Wnt-receiving epithelial stem cells provides self-renewal and regenerative capacity to the colonic epithelium (Degirmenci et al, 2018).

During inflammation, CNs can be perturbed by the induction of aberrant cell states and recruitment of non-resident cell types, much like natural ecosystems are disturbed by the arrival of alien species (Stuart Chapin et al, 2002). Inflammation-induce CNs can facilitate tissue regeneration and reestablishment of homeostasis, or can act as drivers of pathology, especially in chronic settings such as inflammatory bowel diseases (IBD). Understanding how CN architecture is altered by inflammation is fundamental to gaining insights into pathological mechanisms. Most recently, CNs have been inferred from single-cell RNA sequencing (scRNAseq) data by computationally predicting CCIs based on the expression levels of ligands and their receptors (Efremova et al, 2020; Browaeys et al, 2020). Due to tissue dissociation, however, scRNAseq only provides a fragmented view of a tissue ecosystem: cells predicted to interact might populate spatially distinct areas of the tissue, and thus are unlikely to constitute a genuine CN. Hence there is a need for analytic methods that leverage spatial information to shortlist and prioritize CCIs that are more likely to occur in tissues.

Next generation sequencing (NGS)-based spatial transcriptomics (ST) methods, such as Visium ST (10× Genomics), capture RNA molecules in situ at spatially barcoded spots, generating bulk RNA profiles of 10–30 cells (Rao et al, 2021). While the coarse resolution of these methods is generally considered a disadvantage, we show here that these "mixed transcriptomes" can be used to infer CNs, as their gene expression profiles contain information about cell type composition, expressed ligand-receptor (LR) pairs, as well as signaling pathways, gene regulatory networks and effector molecules that mediate CN function. We hypothesize that the inherent spatial proximity of spot data can be leveraged to integrate these three levels of information, reconstructing the architecture and function of CNs. To this end, we developed ISCHIA (Identifying Spatial Co-occurrence in Healthy and InflAmed tissues), a computational framework that assigns a quantitative property to the interaction potential of cell types or LR pairs by computing their spatial co-occurrence within each spot. This

[1]Roche Pharma Research and Early Development, Immunology Infectious Diseases and Ophthalmology Discovery and Translational Area, Grenzacherstrasse 124, 4070 Basel, Switzerland. [2]Department of Biosystems Science and Engineering, ETH Zürich, Mattenstrasse 26, 4058 Basel, Switzerland. [3]These authors contributed equally: Atefeh Lafzi, Costanza Borrelli. ✉E-mail: andreas.moor@bsse.ethz.ch

probabilistic approach is inspired by species co-occurrence models in ecology, which derive statistically significant patterns of pairwise species associations from the frequency of their observed co-occurrence at defined spatial locations (Veech, 2013). Co-occurrence between two species may be positive (the observed co-occurrence is higher than expected by chance), random (independently distributed), or negative (lower than expected by chance). For decades, ecologists have analyzed co-occurrence patterns of plant and animal species to understand ecological communities and the rules of their assembly (Veech, 2013). Here, we apply the same principles to gain deeper insights into cellular communities and the rules of their spatial associations in tissues.

We applied ISCHIA to chart the CN landscape of the healthy and inflamed human colon. We generated both sequencing-based (Visium) and hybridization-based (Molecular Cartography) ST data of human ulcerative colitis samples, and applied ISCHIA to identify pairs of cell types that are co-occurring in inflammation-specific cellular neighborhoods. Within inflammatory neighborhoods, we then performed co-occurrence analysis of ligand and receptor genes, and derived associated gene signatures. Thereby, we reconstructed the architecture of a M cell–fibroblast network enriched in the inflamed colon. We next extended co-occurrence analysis to the whole surfaceome (all ligands and receptor genes) to uncover spatially coordinated tissue responses to inflammation. Finally, we applied co-occurrence analysis to a spatial transcriptomics dataset of murine colitis and identified conserved tissue repair pathways.

## Results

### Composition-based clustering of spots divides tissues into composition classes

We hypothesized that CNs would be best reconstructed within individual spots (intra-spot analysis), as their mixed transcriptome contains information about locally occurring cell types, expressed ligands and receptors, and activated signaling pathways. This intra-spot approach is distinct from state-of-the-art analysis tools for Visium data, which consider each spot as a single datapoint, and compute co-localization, network or cell-cell interactions analysis between neighboring spots (inter-spot analysis).

As inferring CNs in each individual spot separately would be noisy, sparse, computationally intensive, and would lack statistical power, ISCHIA first divides the tissue into clusters of spots with similar cellular composition—termed composition classes (CCs) (Fig. 1A). CCs are thus groups of spots containing similar mixtures of cells, or cellular communities, e.g., all spots capturing colonic crypts. To achieve the division of the tissue into CCs, spot transcriptomes are deconvoluted, yielding a cell type composition matrix (spot × contribution of each cell type, calculated as identity probability $P$), which is then subjected to dimensionality reduction and $k$-means clustering. ISCHIA allows for both reference-based deconvolution, with tools such as SPOTlight (Elosua-Bayes et al, 2021) or RCTD (Cable et al, 2022), and reference-free deconvolution (Miller et al, 2022). Upon deconvolution, ISCHIA summarizes spot gene expression data in a binary presence–absence matrix, where each cell type with $P > 0.1$ is listed for each spot. Each spot is thus represented as a mixture of cell types, and similar mixtures

are clustered together in CCs. Co-occurrence of cell types and transcript species is then calculated within each CC separately. We applied ISCHIA on a publicly available Visium slide of a coronal section of the mouse brain (10× Genomics), using as a reference for deconvolution a scRNAseq dataset of adult mouse cortical cells from the Allen Institute (Tasic et al, 2016). Composition-based clustering of the spots yielded 5 CCs, which broadly reflect the annotated anatomical regions (Fig. EV1A). ISCHIA then computes cell type co-occurrence for every CC, identifying spatial association of cell types (Fig. EV1B). ISCHIA reconstructs cellular networks within spots with cell types as nodes, and is therefore distinct from inter-spot analysis employed by other tools on this same sample, in which spots are used as nodes of the inferred interaction network (Palla et al, 2022; Del Rossi et al, 2022).

To demonstrate the ability of ISCHIA to chart the CN landscape of healthy and diseased tissues, we generated Visium data from 4 inflamed and non-inflamed colon resections of 3 ulcerative colitis (UC) patients (Fig. EV2A). We took advantage of published (Smillie et al, 2019; Martin et al, 2019, Handler et al, 2023) and in house datasets to compile a comprehensive, integrated IBD scRNAseq reference (51 patients in total, Fig. EV2B). Notably, the Handler dataset was collected using the microwell-based platform BD Rhapsody, and thus contains granulocytes that are mostly absent from data generated with droplet-based methods. Using our integrated reference, we performed joint deconvolution of Visium spots across all samples (see Methods, Fig. 1B). Principal component analysis of the deconvolution matrix revealed no association of a particular sample with the first 4 principal components (Fig. EV2C,D). Subsequent dimensionality reduction and k-means clustering of the deconvolution matrix revealed 8 CCs of co-localizing cell types present in all samples (Fig. 1C,D). In the healthy colon, CCs broadly reflected component layers of the colonic wall: submucosa (CC3), crypt bottom (CC6), and crypt top (CC5 and CC1) (Fig. 1E). In the inflamed samples, the spatial boundaries between CCs were lost, indicating perturbed tissue architecture and altered spatial arrangement of cells (Fig. 1E). The most prevalent cell types across all composition classes were transit amplifiers (TAs), colonocytes and goblet cells, reflecting the cellular composition of the adult colonic epithelium (Elmentaite et al, 2021; Fig. 1F). CC1 and CC5 were enriched in the inflamed colon and were therefore termed inflammatory CCs (Fig. 1G). Indeed, their cellular composition reflects the recruitment of macrophages, a hallmark of IBD pathogenesis (Han et al, 2021; Na et al, 2019). These results indicate that subdividing tissue into CCs not only recapitulates tissue morphology and architecture, but is also able to capture alterations in cell type composition across conditions. Of note, CC2, CC4, CC7 and CC8 were not considered for downstream analysis as they mapped onto the muscular layer, or were highly sample-specific (Fig. EV2E,F).

### Cellular co-occurrence in the inflamed colon

We next compared co-occurrence of cell types between inflammatory and homeostatic CCs. Indeed, we hypothesized that inflammatory CCs would contain CNs induced by spatial rearrangement of cells and infiltrating leukocytes. ISCHIA identified several pairwise cell type co-occurrences as being significantly more frequent than expected by chance in inflammatory CCs (positive co-occurrence, $p$ value < 0.05) (Fig. 2A,B). For example, many

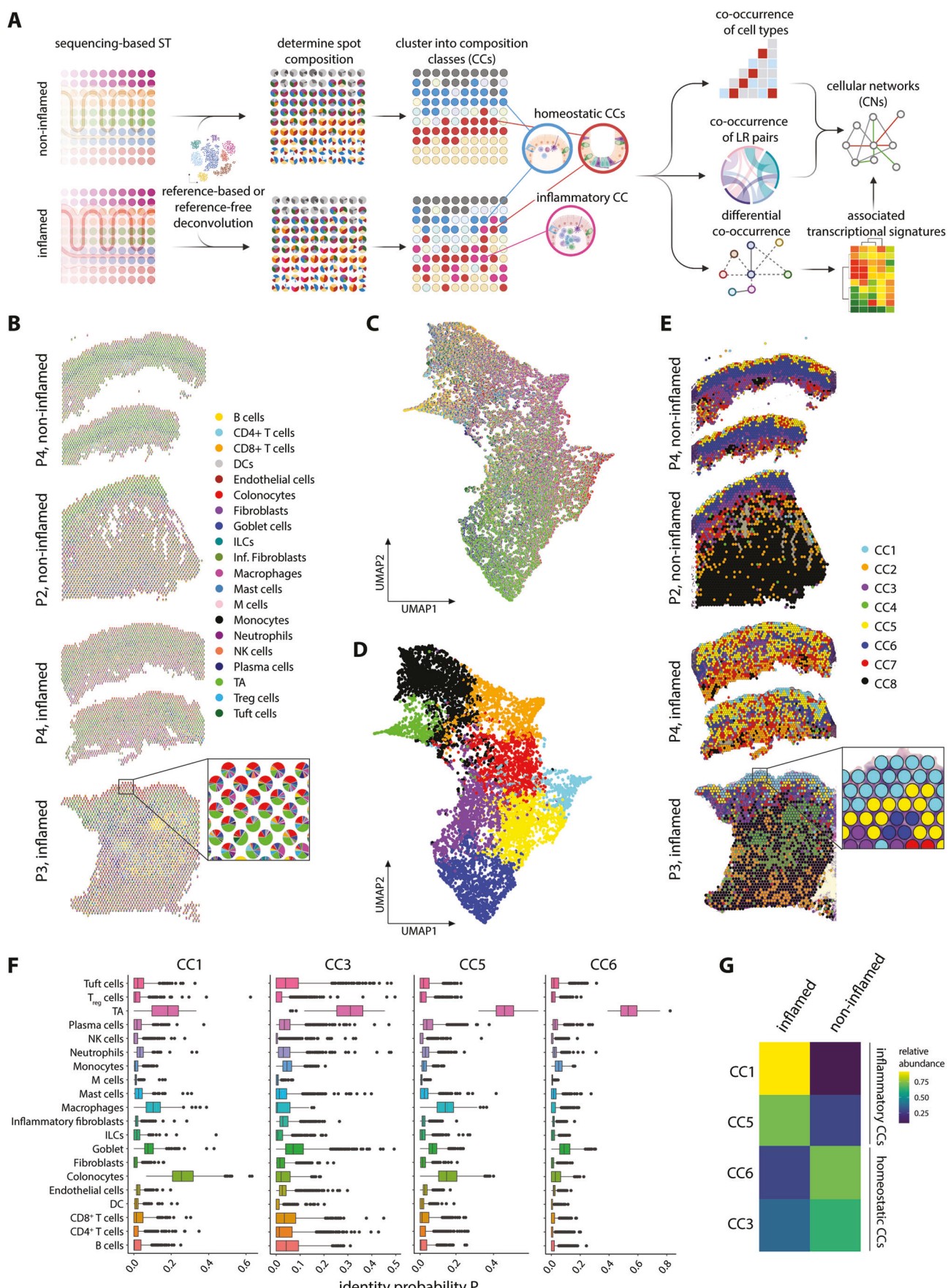

**Figure 1. ISCHIA performs composition-based clustering of spot data.**

(A) Schematic workflow of the ISCHIA pipeline. (B) Deconvolution of Visium spots based on a integrated IBD scRNAseq reference dataset. Spot composition is visualized as a pie chart (shown in inset) and projected on the spatial coordinates. (C) Dimensionality reduction of deconvoluted matrix from all samples ($n = 4$). (D) Composition-based clustering (k-means) of the deconvoluted matrix yields 8 composition classes (CC1–8), representing groups of spots with similar cell type mixtures. (E) Spots overlayed on tissue coordinates, colored by CC. (F) Predicted cell type proportions in 4 CCs comprising the epithelial layer. Boxplots indicate median, first and third quartiles. Whiskers extend from the hinges to the largest value no further than 1.5× the inter-quartile range. Data points beyond the end of the whiskers are plotted individually. $n = 4$ samples from 3 patients. (G) Relative abundance of CCs in inflamed and non-inflamed samples. CC composition class, TA transit amplifiers, ILC innate lymphoid cells, DC dendritic cell.

cellular co-occurrences involving M cells—highly specialized epithelial cells involved in antigen presentation (Dillon and Lo, 2019)—were positively co-occurring in inflammatory CCs but not in homeostatic CCs. Of note, ISCHIA's co-occurrence predictions are irrespective of cell type abundance. Indeed, being the most prominent cell type in all CCs, TAs are co-occurring with many cell types. However, this is not higher than what is expected given their abundance.

We further subdivided spots of inflammatory CCs in those arising from inflamed vs non-inflamed samples, and found that M cell co-occurrences were specifically induced by inflammation (Fig. 2C). Recently, M-like cells were indicated as an interaction hub during colitis based on CCI predictions from scRNAseq data (Smillie et al, 2019). Here, we complement this finding with spatial information and prioritize M cell interactions with monocytes, neutrophils, dendritic cells, and fibroblasts based on the co-occurrence of these cell types specifically in the inflamed colon.

To validate ISCHIA's co-occurrence predictions from spot data (bulk) on a single cell level, we performed 100-plex RNA fluorescent in situ hybridization (FISH, Molecular Cartography). Contrarily to sequencing-based ST methods such as Visium, multiplexed in situ hybridization techniques achieve single-cell resolution. Co-occurrence can thus be calculated in even smaller cellular neighborhoods (e.g., k nearest neighbors = 5 cells), greatly restricting the interaction space and further refining predictions. For this analysis, we included an additional inflamed and non-inflamed sample, for a total of 6 human UC colon samples from 4 patients. Nuclear DAPI staining was used to segment Molecular Cartography data into putative single cells. Segments were then annotated based on marker gene expression and subsequently used as input for cell type co-occurrence analysis. Confirming predictions by ISCHIA based on spot data, Molecular Cartography revealed that M cells (SPIB+ segments) and fibroblasts (PDGFRA/PDPN+ segments) were significantly co-occurring in inflamed but not in non-inflamed samples (Fig. 2D,E). Importantly, the increased M cell and fibroblast co-occurrence was independent of their abundance (Fig. 2F). Thus, ISCHIA has the capacity to capture interactions arising through co-occurrence that has been newly established—possibly by spatial rearrangement or altered distribution of the involved cell types—even if cell type frequencies are unaltered (Fig. 2G).

Co-occurrence analysis can thus be used on both spot data and FISH-based ST data, to infer the interaction potential of cell types based on their proximity. In ecology, species co-occurrence is not generally considered evidence of biotic interactions (Blanchet et al, 2020), however, we argue that it can be used as a measure of spatial association and proximity, prerequisites for juxtacrine and paracrine signaling between cells.

## Linking cell type and ligand–receptor co-occurrence to reconstruct cellular networks

We next applied spatial co-occurrence analysis to transcript species encoding for ligand and receptor genes. We reasoned that the potential for molecular interaction is higher between LR pairs which co-occur in the tissue more frequently than expected by chance. To test this hypothesis, we performed co-occurrence analysis of LR genes within CC5, identified positively co-occurring pairs ($FDR < 0.05$, Dataset EV1), and ranked them based on the correlation of their expression (see "Methods," top 20 shown in Fig. 3A). Several of the significantly co-occurring LR pairs were previously implicated in colitis, and are mostly involved in epithelial integrity and repair (DDR1-CDH1 (Li et al, 2022), DSG2-DSC2 (Gross et al, 2018)), immune recruitment (MIF-CD74 (Farr et al, 2020)), and TNF signaling (GRN-TNFRSF1A (Terryn et al, 2021; Wei et al, 2014)). This highlights ISCHIA's ability to capture disease-relevant pathways from spatial transcriptomics data, even at a limited sample size. Of note, co-occurrence analysis is agnostic to gene expression levels (gene count threshold ≥1) and only takes into account the proximity (measured as co-occurrence) of a given ligand and receptor (see "Methods"). The count threshold is a user defined parameter that can be increased to restrict the co-occurrence analysis to highly expressed ligands and receptors. To account for the sparsity of ST data, ISCHIA calculates LR co-occurrence within CCs (clusters of spots with similar cell mixtures), as aggregating spots in CCs mitigates the effects of low transcript capture rate and consequent false negative predictions.

We hypothesized that positively co-occurring cell types may interact via positively co-occurring LR pairs, and thus focused on M cells and fibroblasts. Using the integrated IBD scRNAseq reference, we identified two ligands expressed by M cells (ADAM15 and VEGFA) with positively co-occurring receptors expressed by fibroblasts (ITGA5 and PDGFRA) (Figs. 3B and EV3A). ISCHIA thus reconstructs nodes and edges of CNs by computing co-occurrence of cell types and LR pairs. Importantly, co-occurrence analysis captures interactions that arise via spatial rearrangement of the LR-expressing cells, even if transcript and cell type abundance are unaltered (Fig. 3C). Indeed, while ITGA5 and ADAM15, and VEGFA and PDGFRA are positively co-occurring in inflammatory CC5, their expression is unaltered by inflammation (Fig. EV3B). Molecular Cartography confirmed co-localizing expression of ITGA5 by PDGFRA/PDPN+ fibroblasts and of ADAM15 by SPIB + M cells (Fig. 3D). Moreover, an independent immunohistochemical analysis of IBD colon sections showed close contact between ADAM15-positive epithelial cells and α5β1(ITGA5)-positive myofibroblasts in regenerative areas (Mosnier et al, 2006). By linking cell type and LR co-occurrence we thus identify a new CN arising in the colon of UC patients (Fig. 3E). While

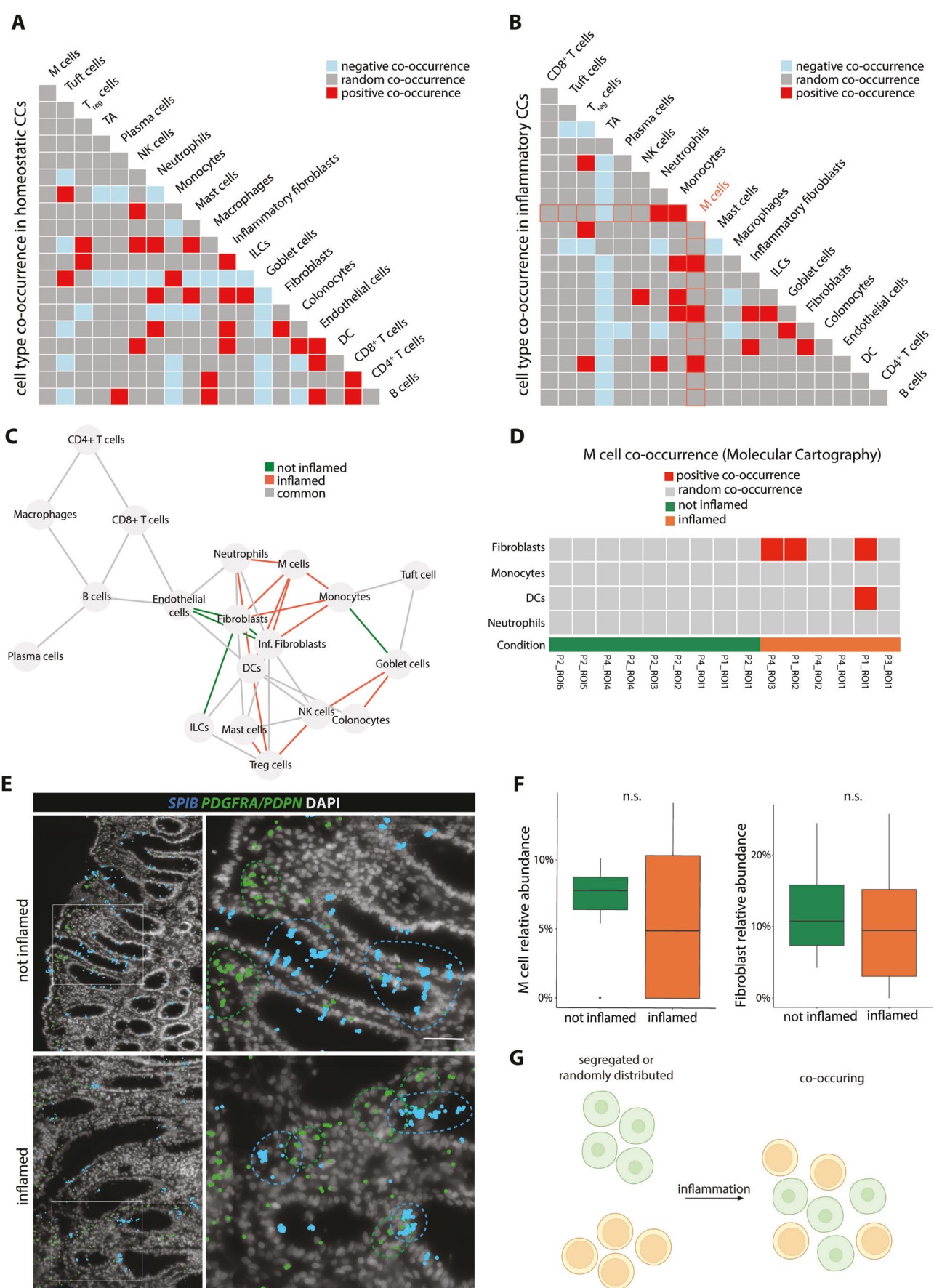

**Figure 2. M cells and fibroblasts co-occur in the inflamed human colon.**

(A, B) Diagonal matrix plot depicting cell type co-occurrences in homeostatic CCs (left) and inflammatory CCs (right). Co-occurrence is positive when observed more frequently than expected ($P < 0.05$), random when there is no significant difference, negative when observed less than expected ($P < 0.05$). Underlying statistical analysis outlined in "Methods." (C) Cell type co-occurrences in inflammatory CCs, shown as a network. Nodes are cell types, edges indicate positive co-occurrences, colored by condition. (D) M cell co-occurrence in spatially restricted cellular neighborhoods ($k = 5$) quantified separately for each ROI from Molecular Cartography data. (E) *SPIB* (M cells) and *PDGFRA/PDPN* (fibroblasts and inflammatory fibroblasts) expression in inflamed and non-inflamed colon as shown by Molecular Cartography. Dashed lines indicate areas of elevated *SPIB* (blue lines) or *PDGFRA/PDPN* (green lines) expression. Scale bar 20 µm. (F) M cell (*SPIB*+ segments) and fibroblasts (*PDGFRA/PDPN*+ segments) relative abundance in 6 UC patient samples, as measured by Molecular Cartography. Boxplots indicate median, first and third quartiles. Whiskers extend from the hinges to the largest value no further than 1.5× the inter-quartile range. Data points beyond the end of the whiskers are plotted individually. $n = 6$ samples from 4 patients. n.s. non significant, unpaired Wilcoxon test. (G) Spatial rearrangement of cells during inflammation leads to positive co-occurrence. Source data are available online for this figure.

functional characterization of this interaction in IBD is outstanding, it illustrates the hypothesis-generating power of co-occurrence analysis performed by ISCHIA.

## Integrating cell types, LR pairs, and transcriptomic signatures to infer CN function

An actively signaling LR pair is likely associated with a specific transcriptional signature, which may be either upstream (inducing LR expression) or downstream (induced by LR signaling). As spot data are only temporal snapshots, we cannot distinguish between causative and consequential effects, but capture the sum of both as a LR-associated transcriptional signature.

After identifying putatively disease-relevant LR pairs, ISCHIA performs differential gene expression analysis between spots of the same CC expressing or lacking that LR pair. Importantly, by doing this within the same CC, transcriptomic effects arising from differences in cell type composition are filtered out, retaining only effects derived from alterations in cell state that are associated with the presence or absence of a given LR pair. Within CC5, we computed significantly differentially expressed genes (DEGs, FDR < 0.05) between spots having or lacking *ITGA5-ADAM15* expression (Fig. 3F). Pathway analysis of the resulting significant DEGs revealed enrichment of terms related to fibronectin matrix formation, as well as signaling events mediated by integrin-linked kinase and focal adhesion kinase (FAK) (Fig. EV3C). These results corroborate recent studies indicating that *ITGA5* is important for fibronectin assembly by fibroblasts (Lu et al, 2019) and that ADAM15 interaction with integrin αV (*ITGA5*) activates FAK signaling to promote migration (Zhou et al, 2022). Of note, these pathways were not enriched in *VEGFA-PDGFRA*-associated gene sets, which instead are involved in cell-junction organization and the HIF-1α transcription factor network (Figs. 3G and EV3D), possibly indicating a coordinated hypoxia-response.

Using spot data from unbiased, genome-wide spatial transcriptomic profiling, cell type co-occurrence and LR co-occurrence can thus be correlated to alterations in the transcriptome, granting deeper insights into the function of reconstructed CNs.

## Differential co-occurrence identifies niche-specific response programs

Ligands and receptors within a spot make up the words of the locally occurring cellular "conversations." We computed the co-occurrence of genes encoding for ligands and receptors within all spots in our dataset (across all CCs), irrespective of whether they are annotated as interacting pairs based on PPI predictions databases. We next selected differentially co-occurring pairs: LR genes that only positively co-occur in a composition class ($FDR_{CCx} < 0.05$ and $FDR_{CCy} > 0.05$) or in a condition ($FDR_{condition\_x} < 0.05$ and $FDR_{condition\_y} > 0.05$). We reasoned that these particular "conversations" may reveal coordinated tissue responses specific to distinct niches or disease-relevant states.

We first analyzed differentially co-occurring LR genes unique to the inflammatory CC5, which is characterized by a high relative abundance of macrophages. We identified differentially co-occurring LR genes that are significantly positively co-occurring in CC5 ($FDR_{CC5} < 0.05$) but not in other CCs ($FDR_{other\_CCs} > 0.05$), indicating a tight association with this particular cell type composition (Fig. 4A). The strongest differentially co-occurring LR network (ranked by $P$ value difference and observed co-occurrence in CC5) involves *EDN1*—an endothelium-derived vasoconstricting peptide implicated in the pathogenesis of IBD (Angerio et al, 2005). Among the 28 *EDN1*-involving LR co-occurrences, *EDN1-CDH17*, *EDN1-CDH1* and *EDN1-F2RL1* are predicted protein-protein interactions. We further detected differential co-occurrence networks specific to CC5 centered around the epithelium-derived pro-angiogenic factors *SEMA3C* (Banu et al, 2006; Yang et al, 2015) and *CXCL5* (Owen and Mohamadzadeh, 2013). Molecular Cartography confirmed localized expression of epithelial *SEMA3C* with *PTPRB* (expressed by the endothelium), *MMP9* (macrophages and monocytes), *CXCL8* (macrophages), *CXCL5* (epithelium, inflammatory fibroblasts, neutrophils) and *SEMA4D* (T cells) (Fig. 4B).

Molecular Cartography further revealed highly spatially restricted and sporadic epithelial expression of *CXCL5* in UC colon samples, which coincided with neutrophil-rich areas, identified by *ITGAM* transcript (Fig. 4C). CXCL5 expression is characteristic of colon adenocarcinoma (Chen et al, 2019; Situ et al, 2022; Lin et al, 2021) and colitis (Keates et al, 1997; Z'Graggen et al, 1997; Rieder et al, 2001), but CXCL5+ cells are also occasionally found scattered in the healthy colon (Z'Graggen et al, 1997), indicating highly localized induction of this cytokine. DEGs associated with differentially co-occurring *CXCL5* interactions (Fig. EV4A) indicate a strong SOX2 and HIF1-α transcriptional signature and enrichment of gene sets associated with innate immunity and neutrophil degranulation (Fig. EV4B). Interestingly, SOX2-induced expression of CXCL5 was previously linked to neutrophil recruitment in non-small-cell lung cancer (Mollaoglu et al, 2018). Whether this signaling axis is involved in IBD is largely

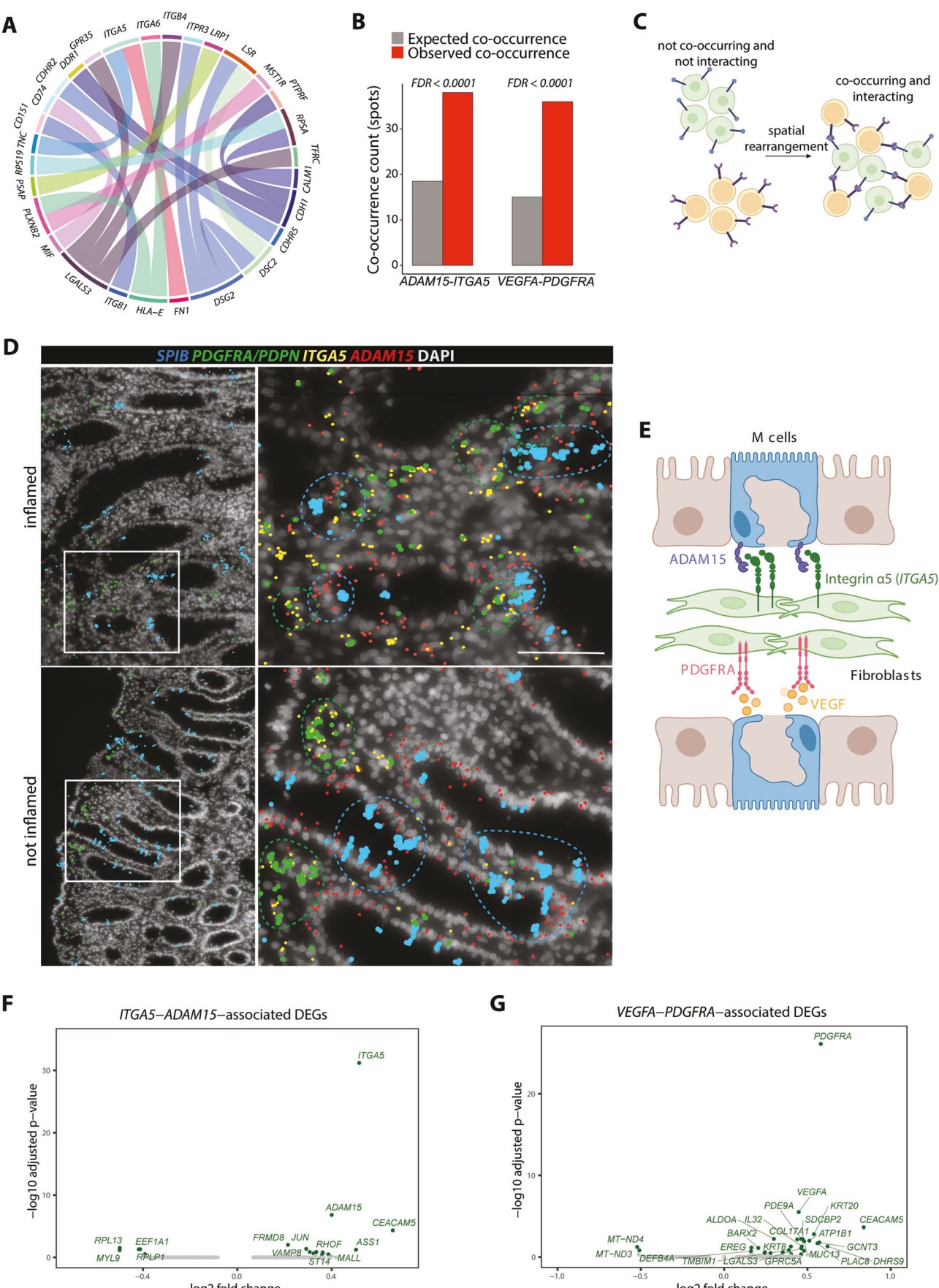

**Figure 3. Reconstructing cellular networks from cell type co-occurrence, LR co-occurrence and corresponding transcriptomic signatures.**

(A) Top 20 positively co-occurring LR pairs in inflammatory CC5 (observed co-occurrence > expected co-occurrence, FDR < 0.05). Ranking based on Spearman correlation of expression of ligand and receptor genes within spots. (B) Expected vs observed co-occurrence (spot count) of *ADAM15* and *ITGA5*, and *VEGFA* and *PDGFRA*. (C) Spatial rearrangement of cells induces positive co-occurrence of LR pairs even if expression levels are unaltered. (D) *SPIB*, *PDGFRA/PDPN*, *ADAM15* and *ITGA5* expression in inflamed and non-inflamed colon as shown by Molecular Cartography. Dashed lines indicate areas of elevated *SPIB* (blue) or *PDGFRA/PDPN* (green) expression. Scale bar 20 μm. (E) Proposed cellular network in inflammatory CCs. (F, G) DEGs associated with *ITGA5-ADAM15* and *VEGFA-PDGFRA* pairs. Non-parametric Wilcoxon rank sum test.

unknown. Moreover, the observation of scattered *CXCL5+* crypts in both inflamed and non-inflamed samples may indicate that aberrant, localized *CXCL5* expression is an early event in UC pathogenesis, and warrants further investigation.

These data suggest that differential co-occurrence analysis of LR genes grants insights into concerted tissue transcriptional responses that only arise in specific cellular neighborhoods. While spatial co-occurrence does not necessarily represent physical binding nor molecular interaction, ligands and their cognate receptors are bound to be co-occurring in close spatial proximity for paracrine and juxtacrine signaling. Whether differential co-occurrence analysis can be used to predict novel putative LR pairs based on their spatial associations in tissues remains to be orthogonally validated.

## Differential co-occurrence reveals an altered crypt surfaceome

Inflammation may not only generate new CNs, but also alter the architecture of existing ones by changing the way the same cell types interact with each other. This would be reflected in altered patterns of positively co-occurring LR pairs. We tested this hypothesis by analyzing the homeostatic CC6, which is present in both inflamed and non-inflamed samples, and maps onto the lower colonic crypt, where epithelial stem cells reside (Fig. 1E–G). We computed differential LR co-occurrence between inflamed and non-inflamed spots of CC6. 9 pair-wise LR combinations were differentially co-occurring in inflamed samples ($FDR_{inflamed} < 0.05$ and $FDR_{non-inflamed} > 0.05$, differential co-occurrence calculated as $FDR_{inflamed} - FDR_{non-inflamed}$) (Fig. 4D). As above, by performing this analysis across conditions but within the same CC, we filter out effects arising from different cell type compositions, and retain the differential interactions of the same groups of cells. Specifically, we detected differential co-occurrence of *Complement 3* (C3) with *MDK* and *SAA1*, all of which have roles in the recruitment of leukocytes (Daffern et al, 1995; Weckbach et al, 2014; De Buck et al, 2018; Davis et al, 2021). In addition, inflammation-specific differential co-occurrence of *C3* and trefoil factor 1 (*TFF1*) might indicate a tissue-protective program, as both factors were implicated in epithelial maintenance (Kulkarni et al, 2019; Hoffmann, 2020).

The differentially co-occurring LR network centered around *SECTM1* (Secreted And Transmembrane 1) similarly suggests a balance between immune cell recruitment and tissue protection. SECMT1 is a soluble protein highly expressed by crypt-top colonocytes (mostly in CC1), that exerts immunomodulatory and chemotactic functions via CD7 on T cells (Wang et al, 2012; Huyton et al, 2011; Wang et al, 2014). In CC6 (crypt bottom), we observed its expression mostly in DCs (*CLEC10A+*) and neutrophils (*ITGAM+*) (Fig. EV4C) but its expression was also recently

reported in intestinal epithelial stem cells (Biton et al, 2018). Interestingly, among DEGs associated with *SECTM1*-interactions are several *CEACAM* genes, encoding for epithelial-derived adhesion proteins involved in immune modulation and colitis (Gray-Owen and Blumberg, 2006; Kelleher et al, 2019; Fig. 4E). Collectively, *SECTM1*-interactions-associated DEGs are enriched in genes involved in the assembly of hemidesmosomes, whose integrity protects the epithelium against inflammation (De Arcangelis et al, 2017; Fig. EV4D). This data suggests that in inflamed colon areas, the crypt epithelium collectively responds to insults by secreting tissue protective factors and by enhancing cell adhesion. Interestingly, *SECTM1* is differentially expressed between pediatric responders and nonresponders to anti-TNF therapy for IBD (Salvador-Martín et al, 2021). However, while the immunomodulatory function of this protein in cancer has been the focus of several studies (Wang et al, 2008, 2012, 2014), its role in colitis is largely unknown.

## Repair pathways are conserved across species

Finally, we applied co-occurrence analysis to an independent Visium dataset of the healing mouse colon (Parigi et al, 2022; Fig. 5A). As no matched scRNAseq dataset was available, we deconvoluted spots in a reference-free manner using topic modeling (see "Methods"), and clustered the deconvoluted matrix in 8 CCs (Fig. 5B). In both the human and murine dataset, we identified LR pairs that are differentially co-occurring in inflammatory vs homeostatic CCs ($FDR_{inflammatory\_CCs} < 0.05$ and $FDR_{homeostatic\_CCs} > 0.05$). Interestingly, we found several orthologous LR pairs that were differentially co-occurring in both mouse and human inflammatory CCs, and whose molecular interaction is also reported in PPI databases (Fig. 5C). Several of those pairs involve *TNFRSF1B*, encoding for the tumor necrosis factor receptor 2, and a genetic risk factor for colitis. Indeed, polymorphisms in *TNFRSF1B* are associated with increased susceptibility to IBD (Ferguson et al, 2009; Sashio et al, 2002; Nagaishi et al, 2016). However, little is known about the function of this gene in inflammation (Punit et al, 2015). Thanks to our comprehensive IBD scRNAseq reference, we could map the expression of *TNFRSF1B* in humans to DCs, monocytes, macrophages and, most prominently, neutrophils (Fig. 5D). Recently, TLR2 signaling in colitis was linked to increased extracellular trap formation by neutrophils (Neuenfeldt et al, 2022), which are emerging as key cellular players IBD (Wéra et al, 2016). However, to what extent this pathway contributes to tissue damage during IBD is still unknown, and highlights the need for cellular atlases that include granulocytes.

Collectively, our comparative analysis in human and mouse colitis suggests that repair CNs are shared across species, and illustrates the ability of ISCHIA to identify co-occurring LR pairs from ST data also in the absence of a single-cell reference.

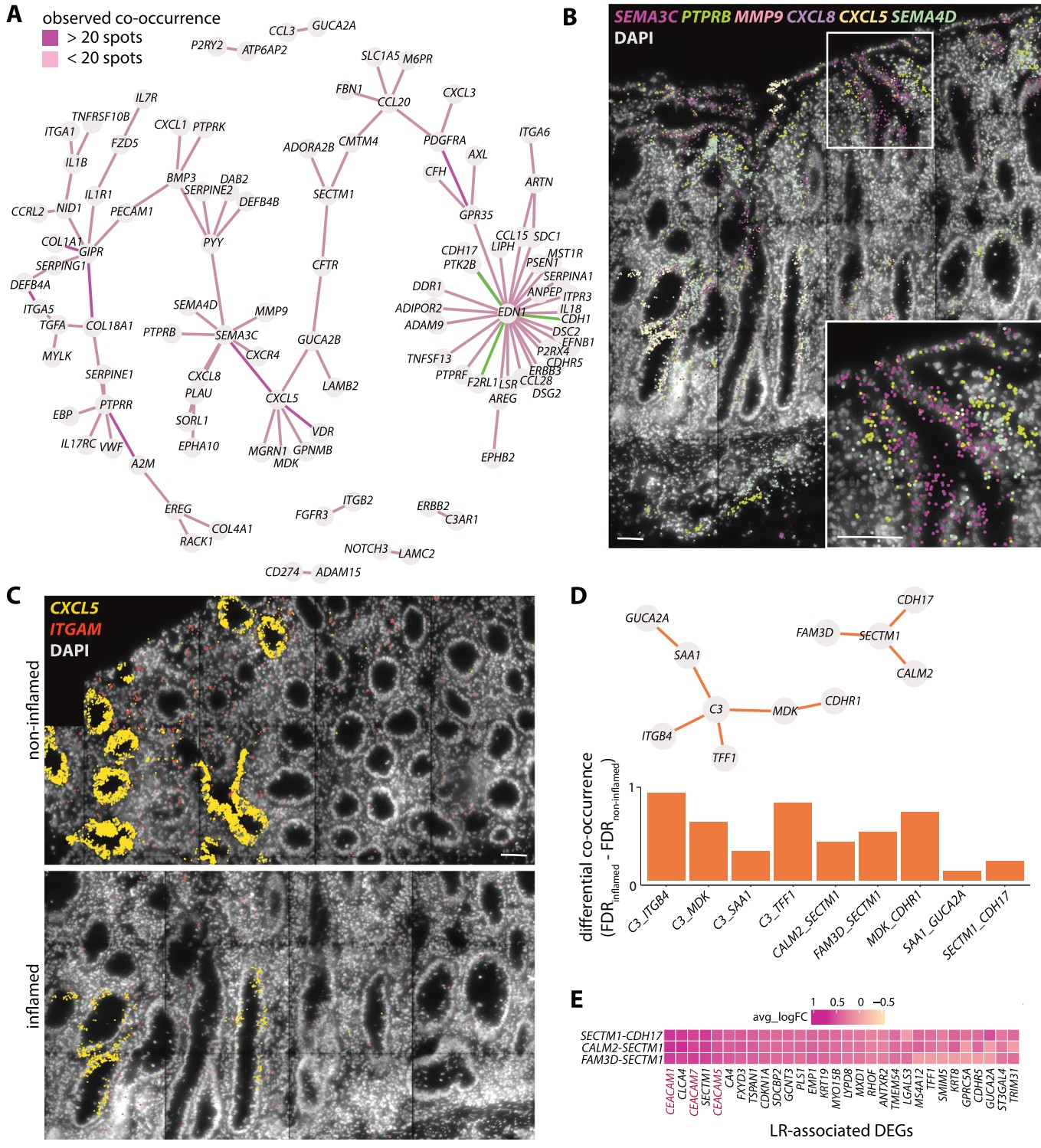

**Figure 4. Differential co-occurrence identifies niche and condition-specific interactomes.**

(A) Network representation of the CC5-specific differentially co-occurring interactome. Purple and pink edges indicate pairwise co-occurrences in more or less than 20 spots, respectively. Green edges indicate pairwise co-occurrences between LRs predicted to interact by PPI databases NicheNet (Browaeys et al, 2020), OmniPath (Türei et al, 2016) or CellTalkDB (Shao et al, 2021). (B) *SEMA3C* co-occurrence network visualized by Molecular Cartography. Scale bar, 20 μm. (C) *CXCL5* and *ITGAM* expression in non-inflamed and inflamed colon samples. Scale bar, 20 μm. (D) Top, Network representation of differentially co-occurring LR genes within inflamed CC6 spots. Bottom, differential co-occurrence score calculated as $FDR_{inflamed} - FDR_{non-inflamed}$. (E) Heatmap of *SECTM1*-interaction-associated DEGs.

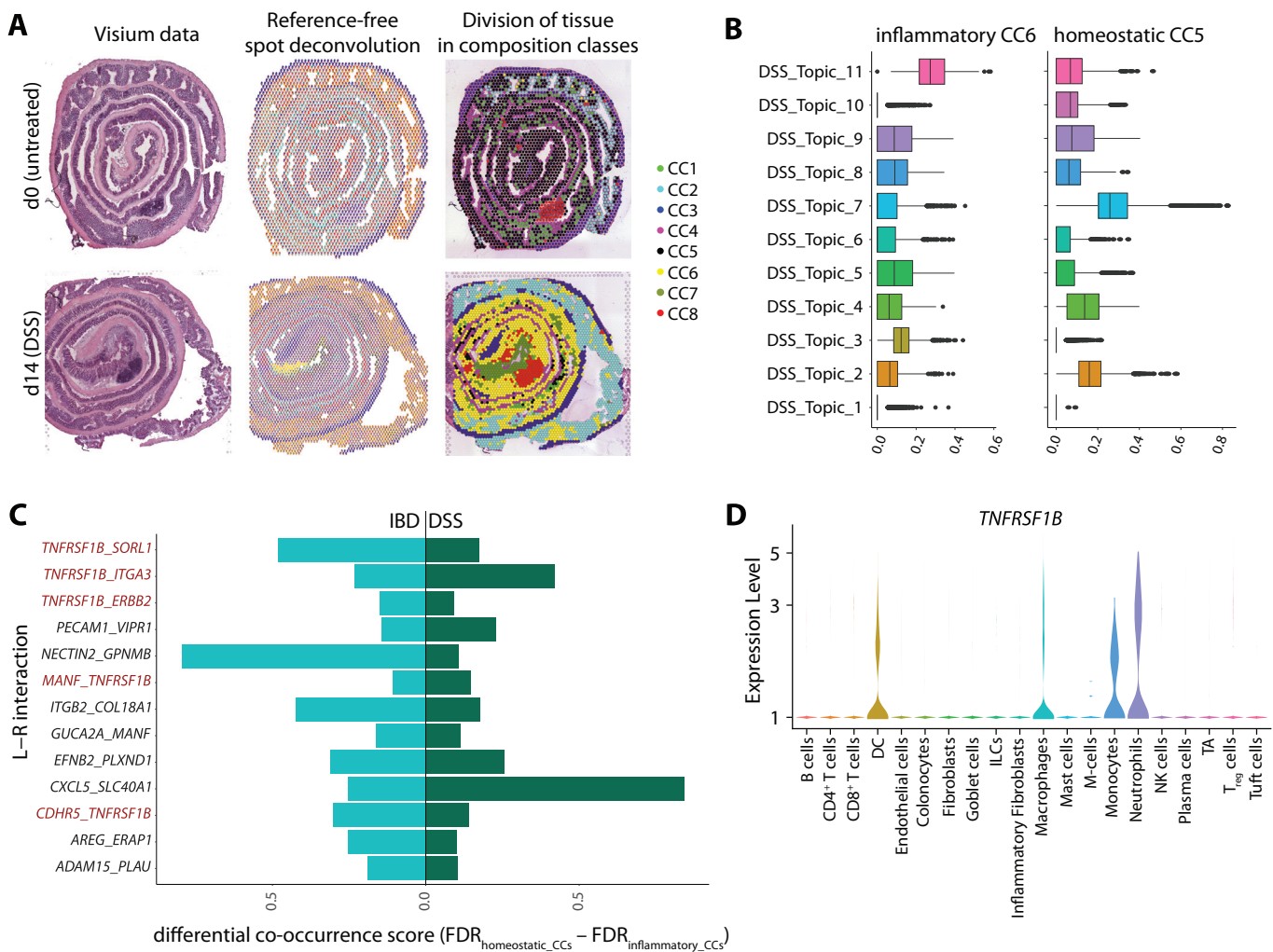

**Figure 5. ISCHIA identifies conserved co-occurring LR pairs.**

(A) Visium data from Parigi et al (2022), representing untreated (d0) and regenerating (d14 post DSS) mouse colon. Left: hematoxylin–eosin staining, center: reference-free spot deconvolution, right: division of tissue in composition classes. (B) Topic distribution in inflammatory and homeostatic CCs in mouse colon. Boxplots indicate median, first and third quartiles. Whiskers extend from the hinges to the largest value no further than 1.5× the inter-quartile range. Data points beyond the end of the whiskers are plotted individually. $n = 2$ samples. (C) Common differentially co-occurring LR pairs with $FDR_{inflammatory\_CCs} < 0.05$ and $FDR_{homeostatic\_CCs} > 0.05$ in both human IBD (left) and mouse DSS (right) Visium data. X-axis shows the differential co-occurrence score, calculated as $FDR_{homeostatic\_CCs} - FDR_{inflammatory\_CCs}$. LR pairs involving *TNFRSF1B* shown in red. (D) Expression of *TNFRSF1B* in human IBD scRNAseq reference. $n = 51$ patients.

## Discussion

In community ecology, co-occurrence analysis is the study of distributions of species at defined spatial locations (Veech, 2013). While the link between co-occurrence and biotic interactions is still debated, modeling of presence–absence data has been instrumental in understanding the rules of assembly of ecological communities (Blanchet et al, 2020). Here, we propose the use of co-occurrence analysis for spatial transcriptomics data. We calculated spatial co-occurrence of cell types and LR genes in Visium data of human UC colon resections. Thereby, we identified a M cell–fibroblast network in inflamed regions of UC patients, which we validated at single-cell resolution by means of Molecular Cartography. We further showed that co-occurrence analysis can successfully be applied to hybridization-based ST data, further refining predictions from

Visium data by analyzing small cellular networks of, for example, $k = 5$ neighboring cells.

ISCHIA differs from other analysis tools for Visium data in that it predicts CNs within spots and not across spots. As proximity is a prerequisite for juxtacrine and paracrine cell-cell communication, which in turn constitutes the basis for the coordinated function of CNs, we hypothesized that CNs would best be reconstructed within individual spots, rather than across neighboring spots. To increase robustness, spots are grouped in clusters of similar cellular composition, termed composition classes. Composition-based clustering of the tissue represents a major advantage of this method, and distinguishes it from other methods, such as Squidpy (Palla et al, 2022) or Giotto (Del Rossi et al, 2022), that assign an identity to each spot based on marker gene expression or on the most abundant cell type. In addition to preserving the complexity of the cell type composition of the analyzed tissue,

composition-based clustering of spots also confers robustness towards variations in expression levels due to batch effects. Indeed, other spatial analysis methods such as Starfysh (He et al, 2022) have found that identifying inter-sample commonalities using composition-based clusters is easier than identifying common transcriptome-based clusters between samples. Still, batch analysis and, if needed, correction of the ST data is recommended prior to analysis with ISCHIA. Composition-based clustering of spots further allows to restrict downstream analysis to similar mixtures of cells, filtering out transcriptome heterogeneity arising from distinct cellular compositions, which might act as a confounder variable when performing differential gene expression or cell-cell interaction predictions.

To reconstruct cellular neighborhoods, ISCHIA performs co-occurrence analysis of cell types within CCs, leveraging the inherent proximity of mixed transcriptomes within individual spots. Hence, the cell types within the spots, rather than the spots themselves, are the nodes of the CN. This is distinct from other tools which build a neighborhood graph using spatial coordinates of spots and a fixed number of adjacent spots (Del Rossi et al, 2022; He et al, 2022; Palla et al, 2022), thereby ignoring the missing data between spots as well as the multicellular nature of each spot. ISCHIA's approach allows for reconstruction of much smaller CNs, operating in close spatial proximity, a prerequisite for juxtacrine and paracrine signaling between cells. ISCHIA further predicts LR interaction as edges connecting cell types within spots, not across spots. Finally, by integrating co-occurrence of cell types, co-occurrence of LR pairs, and associated gene signatures, ISCHIA infers CN function.

Finally, we propose differential co-occurrence analysis of LR genes as a predictive tool to infer concerted and spatially restricted tissue responses. These can be envisioned as "conversations" between cells in a CN, which may vary according to the composition of the CN, or the environmental condition the CN is in (e.g., inflamed vs non-inflamed colon). We define differential co-occurrence as a condition- or niche-specific positive co-occurrence between LR genes. Whether spatial proximity and differential co-occurrence of LR genes can be employed, in conjunction with protein-protein interaction predictions, for the identification of novel putative interacting pairs, warrants further investigation.

Collectively, our study shows that co-occurrence analysis can be applied to both sequencing- and imaging-based ST data, to refine interaction predictions from scRNAseq-based analysis. Co-occurrence analysis is complementary to differential gene expression, as it does not depend on the abundance of a given cell type or transcript, but rather on their spatial in the tissue. Co-occurrence analysis on spatial transcriptomics data can be used to chart the distribution and infer the interactions of cell types and transcripts, revealing disease-specific cellular communities, and predicting juxtacrine and paracrine signaling. It therefore represents a powerful tool for hypothesis-generation from spatial transcriptomics data.

# Methods

## Data analyzed in the study

### scRNAseq

scRNAseq data from IBD patients was obtained from published datasets (Smillie et al, 2019; Martin et al, 2019; Handler et al, 2023) or collected in house with 10× Genomics according to the manufacturer's instructions (2 UC patients, 2 suspected IBD patients, 4 healthy controls).

### 10× Visium

Fresh frozen UC colon samples were sectioned onto a 10× Visium Spatial Gene expression slide. cDNA libraries were generated according to the manufacturer's instructions. After methanol fixation, tissue morphology was assessed by hematoxylin and eosin (H&E) staining. The permeabilization time of 50 min was assessed with the Tissue Optimization Protocol. Lysis, reverse transcription, second strand synthesis and cDNA denaturation were performed on the slide. cDNA was then amplified by PCR using the cycle number identified by qPCR and then subjected to end repair, A-tailing, adapter ligation and indexing to generate sequencing libraries. Quality and quantity of all libraries were assessed using the dsDNA high-sensitivity (HS) kit (Life Technologies #Q32854) on a Qubit 4 fluorometer (Thermo Fisher) and high sensitivity D1000 reagents and tapes (Agilent #5067-5585, #5067-5584) on a TapeStation 4200 system (Agilent Technologies). Paired-end sequencing was performed on a NovaSeq 6000 system (Illumina) using NovaSeq SP Reagent Kits (100 cycles) v1.5. Data was pre-processed using Space Ranger (v1.2.0) (10× Genomics) with GRCm38 v2020-A genecode. Published Visium data were obtained from 10× Genomics (mouse brain) and (Parigi et al, 2022, DSS-treated murine colon).

### Molecular Cartography

Fresh frozen UC colon samples were sectioned onto coverslips and processed by Resolve Biosciences. Cellpose v. 2.0.4 (Stringer et al, 2021) was used to segment nuclei in the DAPI images with the pretrained nuclei model and flow_threshold 0.5, cellprob_threshold -0.2. Using the "expand_labels" function in scikit-image, the nuclear segments were then expanded by 10 pixels (1.38 µm) and transcripts were subsequently assigned to the expanded segments. Segments with less than 3 molecules or 3 genes detected were removed from the analysis.

## Integration and annotation of scRNAseq data

For every scRNAseq dataset, we performed log normalization with scale factor 10,000. The top 2000 variable genes were selected for each dataset using the "vst" method in Seurat (Butler et al, 2018). Next, we used the *FindIntegrationAnchors* function to align shared cellular populations across datasets by finding pairs of cells that are in matching states. The identified anchors are then used in the *IntegrateData* function to calculate an integrated (batch-corrected) expression matrix for all cells, enabling joint analysis. We used the labels provided by (Smillie et al, 2019) as reference for automatic annotation of all the clusters in the integrated data with the *TransferData* function from Seurat. We then manually grouped the clusters into 20 major cell types (Fig. EV3A). Granulocytes were directly retrieved from the Handler et al (2023) dataset.

## Deconvolution of Visium spots

### Reference-based deconvolution

We used the integrated scRNAseq reference to deconvolute human IBD Visium spot data with SPOTlight (Elosua-Bayes et al, 2021), a computational tool that enables the integration of ST and scRNA-

seq data using a seeded non-negative matrix factorization (NMF) regression. The deconvolution function was run with default parameters. The brain coronal Visium sample was deconvoluted with RCTD (Cable et al, 2022) with default parameters.

### Reference-free deconvolution

Mouse colitis Visium data was deconvoluted in a reference-free manner with STdeconvolute (Miller et al, 2022), which builds on latent Drichlet allocation (LDA). Given a count matrix of gene expression in multicellular Visium spots, STdeconvolute applies LDA to infer the putative transcriptional profile for each cell type and the proportional representation of each cell type in each multi-cellular spot. The deconvolution function was run with default parameters.

### Co-occurrence analysis on Visium data

ISCHIA uses spatial co-occurrence, a probabilistic approach inspired by species co-occurrence models in ecology that assigns a measurable property for spatial proximity between cell types or transcripts. Observed co-occurrence is quantified as the number of spatial spots where two cell types or genes co-occur. Observed co-occurrence is compared to the expected co-occurrence, where the latter is the product of the two cell types' probability of occurrence, multiplied by the number of spots: $E\,(N\,cell\,type\,1, 2) = P\,(cell\,type\,1) \times P\,(cell\,type\,2) \times N$.

This probabilistic model uses combinatorics to determine whether the observed frequency of co-occurrence is significantly greater than expected (positive association), significantly smaller than expected (negative association), or not significantly different than expected (random association). Specifically, this analysis calculates, for each cell type or transcript pair, the exact probability of the observed co-occurrence to be greater than ($P(gt)$) or less than ($P(lt)$) the expected co-occurrence. Importantly, this analysis is distribution-free and the results can be interpreted and reported as $P$ values, without reference to a statistic. Therefore, given two cell types in a dataset, a $P(lt) \le \alpha$ suggests that those two species are negatively associated (where $P(lt) = \$p\_lt$ and $\alpha = 0.05$).

To define the cell type composition of each spot, the spot transcriptional profile is deconvoluted. The resulting spot cell type/topic probability matrix is then converted to a binary presence–absence matrix by thresholding (probability P > 0.1 = presence). For LR pairs, the presence–absence matrix is derived by their expression. This matrix is then used as an input for co-occurrence calculation with the R package cooccur (Griffith et al, 2016): we calculate the probability of selecting a spot that has cell type #1 given that it already has cell type #2.

The probability that the two cell types co-occur at exactly j number of spots is given by

$$P_j = \frac{\binom{N_1}{j} \times \binom{N - N_1}{N_2 - j}}{\binom{N}{N_2}}$$

For $j = 1$ to $N_1$ spots: $N_1$ = number of spots where cell type #1 occurs; $N_2$ = number of spots where cell type #2 occurs; $N$ = total number of spots sampled (where both cell types could occur).

The term, $\binom{N_1}{j}$ represents the number of ways of selecting $j$ spots that have cell type #1 given that there are $N_1$ such spots in the "population" of all spots.

The term $\binom{N - N_1}{N_2 - j}$ represents the number of ways of selecting $N_2 - j$ spots that have cell type #2 but not cell type #1 given that there are $N - N_1$ such spots.

The numerator $\binom{N_1}{j} \times \binom{N - N_1}{N_2 - j}$ gives the total number of ways of selecting $j$ spots that have cell types #1 and #2. The denominator $\binom{N}{N_2}$ represents the total number of ways that $N_2$ number of spots could be obtained out of a total of $N$ spots. Thus the equation is giving the proportion of the $N_2$ spots that also have cell types #1 under the condition that the two cell types co-occur at $j$ spots.

The spatial co-occurrence for LR pairs follows the same principle (presence–absence matrix based on count >1). Additionally, ISCHIA ranks LR pairs by calculating the Spearman correlation of expression between the ligand and receptor gene in each spatial spot, expecting that if a LR pair is interacting, the expression of the ligand and receptor should correlate as well. For the calculation of LR-associated DEGs, ISCHIA computes differential gene expression between spots that are double positive or double negative for a given LR pair. The significant DEGs are then used for pathway enrichment with any tool of choice, such as EnrichR (https://bio.tools/enrichr). We employed a permutation-based approach to assess the significance of the obtained DEGs. Specifically, we generated a null distribution of DEGs (noise estimation) by 1000-fold random sampling of spots into two groups, and calculating DEGs between these groups. Next FDR-adjusted Monte Carlo $P$ values were calculated for each LR-associated DEG, comparing the initially computed $P$ values DEG from with the null distribution, and subsequently adjusting for multiple testing. DEGs with Monte Carlo FDR < 0.05 are likely specific to the presence of a given LR and unlikely to result from random spot sampling.

### Co-occurrence analysis on Molecular Cartography data

We applied co-occurrence analysis on Molecular Cartography data by transforming the $x,y$ coordinates of the segmented cells into cellular neighborhoods. To do so, we calculated the $K$-nearest neighbor graph of the $x,y$ coordinates where $k = 5$. We applied ISCHIA's co-occurrence analysis on the presence–absence matrix of calculated CNs and cell types as explained above.

## Data availability

The datasets and computer code produced in this study are available in the following databases: Spatial transcriptomics data: Zenodo repository https://doi.org/10.5281/zenodo.7589581. Code: ati-lz/ISCHIA: Framework for analysis of cell-types and Ligand-Receptor cooccurrences (github.com).

## Peer review information

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

## Acknowledgements

We thank members of the Moor lab and the Roche Immunology Discovery department for insightful discussions. We are grateful to the Basel Genomics facility for technical support. We also thank D. Eletto for organizing our lab retreat on the island of Ischia, where this project was conceived. This project was funded by the Swiss National Science Foundation (PCEFP3_181249). Schematics were created using a licenced version of Biorender.com.

## Author contributions

**Atefeh Lafzi**: Conceptualization; Data curation; Software; Formal analysis; Validation; Investigation; Visualization; Methodology; Writing—original draft. **Costanza Borrelli**: Conceptualization; Data curation; Formal analysis; Validation; Investigation; Visualization; Methodology; Writing—original draft; Writing—review and editing. **Simona Baghai Sain**: Formal analysis; Investigation. **Karsten Bach**: Formal analysis; Investigation. **Jonas A Kretz**: Investigation. **Kristina Handler**: Investigation. **Daniel Regan-Komito**:

Investigation. **Xenia Ficht**: Writing—review and editing. **Andreas Frei**: Resources; Supervision; Project administration. **Andreas Moor**: Conceptualization; Resources; Supervision; Funding acquisition; Project administration; Writing—review and editing.

## Disclosure and competing interests statement
The authors declare no competing interests.

# Expanded View Figures

**Figure EV1. Composition-aware clustering and cell type co-occurrence in mouse brain Visium data.**

(**A**) A Visium sample of a mouse brain coronal section (10× Genomics) is deconvoluted using a scRNAseq reference (Tasic et al, 2016), yielding 5 composition classes. Scale bar, 1 mm. (**B**) Diagonal matrix plot depicting cell type co-occurrences in every CC. Co-occurrence is positive when observed more frequently than expected ($P < 0.05$), random when there is no significant difference, negative when observed less than expected ($P < 0.05$). Underlying statistical analysis outlined in "Methods."

a

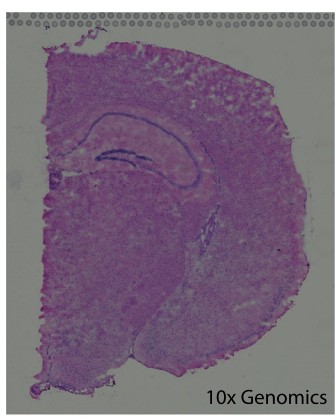

reference-based deconvolution
and
composition-based clustering

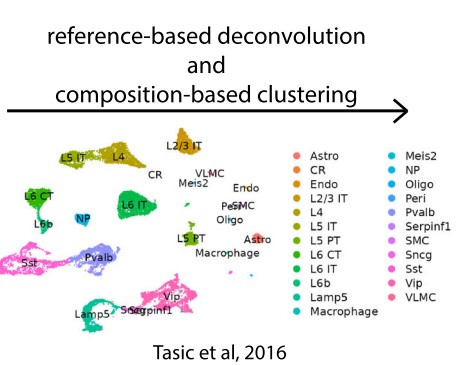

Tasic et al, 2016

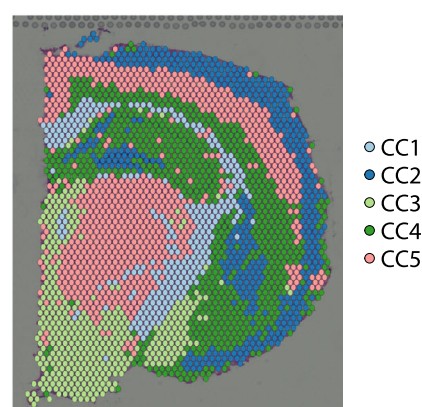

b

### co-occurrence of cell types in CC1

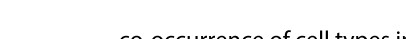
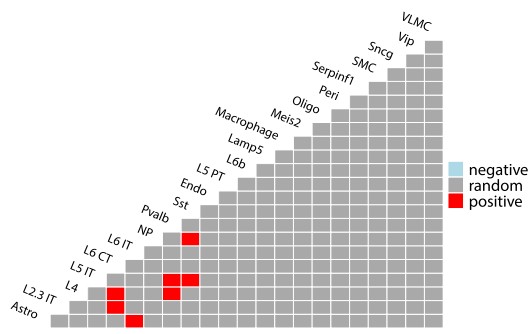

### co-occurrence of cell types in CC2

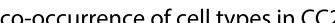
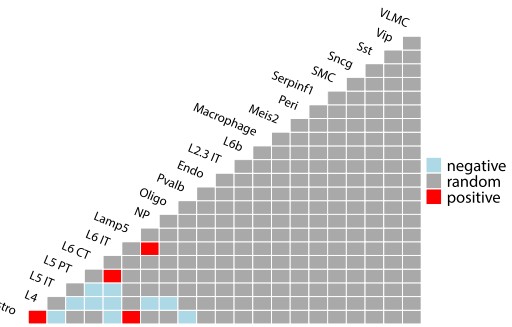

### co-occurrence of cell types in CC3

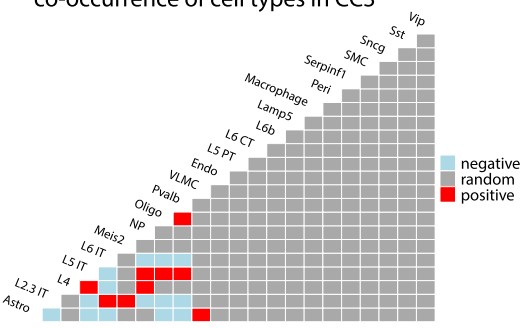

### co-occurrence of cell types in CC4

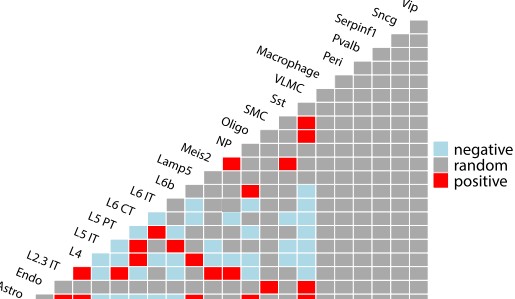

### co-occurrence of cell types in CC5

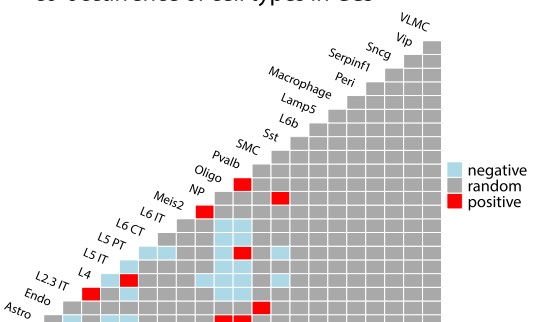

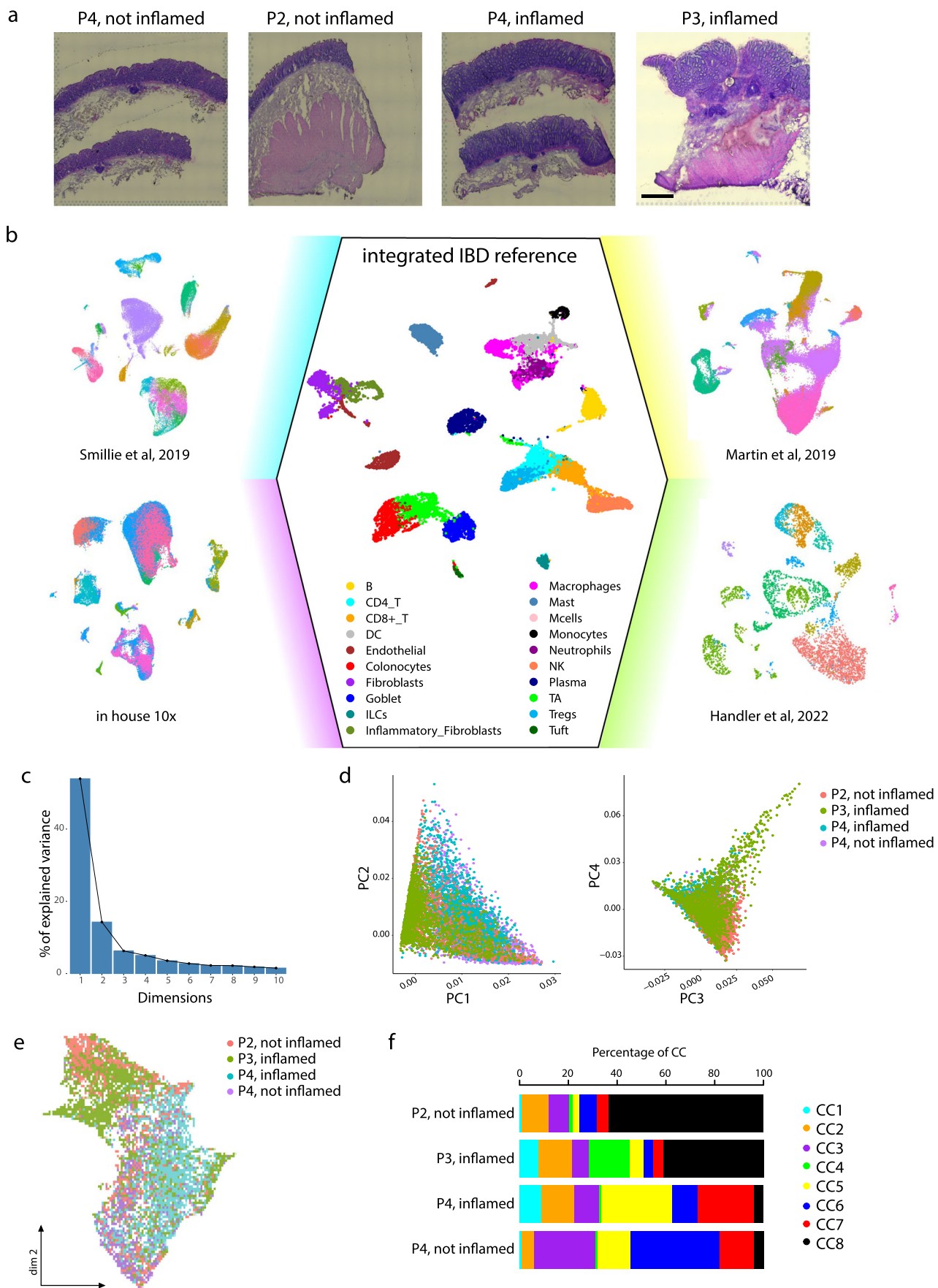

**Figure EV2.  Composition-aware clustering of human colon Visium data.**

(A) Hematoxylin–eosin staining of colon resections from four samples (a total of 3 ulcerative colitis patients, 2 inflamed and 2 not inflamed samples) analyzed by Visium ST (10× Genomics). Scale bar, 1 mm. (B) Integration of published and in house scRNAseq datasets yields a comprehensive IBD reference with a total of 51 patients. Dots represent single cells, colored by cell type. (C) Percent of variance of the deconvolution matrix explained by the first 10 principal components. PCs 1–4 explain more than 80% of the variance. (D) PC plot showing no association of any sample with a particular principal component. (E) Dimensionality reduction of spots colored by sample. (F) Percentage of CC per sample.

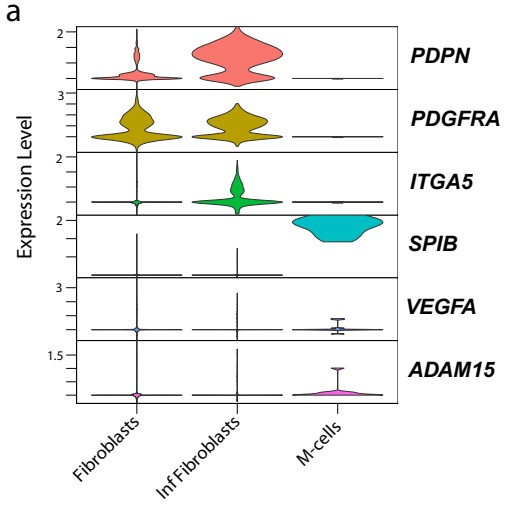

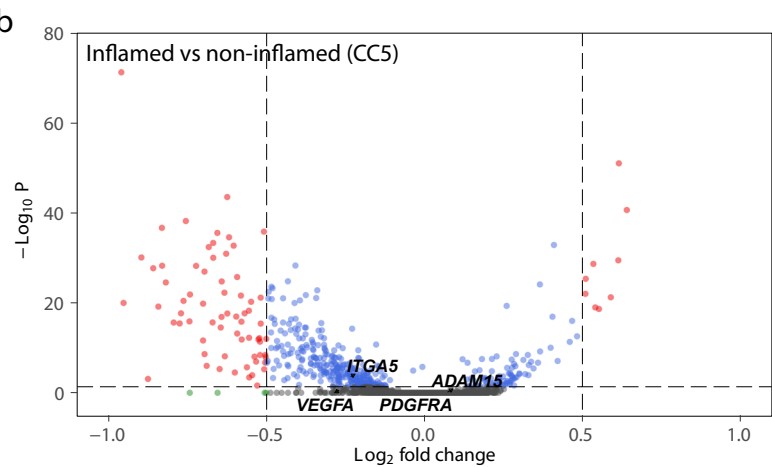

**c** Reactome

Fibronectin Matrix Formation R-HSA-1566977

Peptide Chain Elongation R-HSA-156902

Neutrophil Degranulation R-HSA-6798695

Eukaryotic Translation Elongation R-HSA-156842

Cell Surface Interactions At Vascular Wall R-HSA-202733

Innate Immune System R-HSA-168249

Developmental Biology R-HSA-1266738

Major Pathway Of rRNA Processing In Nucleolus And Cytosol R-HSA-6791226

rRNA Processing In Nucleus And Cytosol R-HSA-8868773

rRNA Processing R-HSA-72312

NCI Nature Pathway Interaction Database

Integrin-linked kinase signaling Homo sapiens 21738158-6194-11e5-8ac5-06603eb7f303

Signaling events mediated by focal adhesion kinase Homo sapiens 8fb80085-6195-11e5-8ac5-06603eb7f303

JNK signaling in the CD4+ TCR pathway Homo sapiens 400ebdab-6194-11e5-8ac5-06603eb7f303

PDGFR-alpha signaling pathway Homo sapiens c66cc833-6194-11e5-8ac5-06603eb7f303

Plexin-D1 Signaling Homo sapiens e3068f36-6194-11e5-8ac5-06603eb7f303

S1P2 pathway Homo sapiens 7796a240-6195-11e5-8ac5-06603eb7f303

Alpha9 beta1 integrin signaling events Homo sapiens 0f5519cf-6188-11e5-8ac5-06603eb7f303

VEGFR3 signaling in lymphatic endothelium Homo sapiens 9048d98c-6196-11e5-8ac5-06603eb7f303

Integrin family cell surface interactions Homo sapiens 1ca2bf67-6194-11e5-8ac5-06603eb7f303

Calcium signaling in the CD4+ TCR pathway Homo sapiens 5294f70b-618f-11e5-8ac5-06603eb7f303

**d** Reactome

Cell Junction Organization R-HSA-446728

Hemostasis R-HSA-109582

Tight Junction Interactions R-HSA-420029

Cell-Cell Communication R-HSA-1500931

Cell Surface Interactions At Vascular Wall R-HSA-202733

O-linked Glycosylation Of Mucins R-HSA-913709

Cell-cell Junction Organization R-HSA-421270

Extra-nuclear Estrogen Signaling R-HSA-9009391

Biosynthesis Of E-series 18(S)-resolvins R-HSA-9018896

Constitutive Signaling By Aberrant PI3K In Cancer R-HSA-2219530

NCI Nature Pathway Interaction Database

HIF-1-alpha transcription factor network Homo sapiens 20ef2b81-6193-11e5-8ac5-06603eb7f303

Integrins in angiogenesis Homo sapiens 2ddeac89-6194-11e5-8ac5-06603eb7f303

PDGF receptor signaling network Homo sapiens c5bcd922-6194-11e5-8ac5-06603eb7f303

VEGF and VEGFR signaling network Homo sapiens 8957818a-6196-11e5-8ac5-06603eb7f303

ErbB receptor signaling network Homo sapiens 2c26d51f-6192-11e5-8ac5-06603eb7f303

S1P1 pathway Homo sapiens 7327884f-6195-11e5-8ac5-06603eb7f303

PDGFR-alpha signaling pathway Homo sapiens c66cc833-6194-11e5-8ac5-06603eb7f303

Plexin-D1 Signaling Homo sapiens e3068f36-6194-11e5-8ac5-06603eb7f303

Alpha9 beta1 integrin signaling events Homo sapiens 0f5519cf-6188-11e5-8ac5-06603eb7f303

VEGFR1 specific signals Homo sapiens 8b13143b-6196-11e5-8ac5-06603eb7f303

**Figure EV3.   Co-occurring LR pairs are associated with a transcriptional signature.**

(A) Expression of cell type markers (*PDPN, PDGFRA, SPIB*) and LR pairs (*PDGFRA, VEGFA, ITGA5, ADAM15*) by fibroblasts and M cells in integrated IBD scRNAseq reference. (B) Differential gene expression within CC5, inflamed vs non-inflamed (4 samples, 3 patients). Non-parametric Wilcoxon rank sum test. (C) Reactome and NCI Nature Interaction Database pathway enrichment of *ITGA5-ADAM15*-associated genes. (D) Reactome and NCI Nature Interaction Database pathway enrichment of *VEGFA-PDGFRA*-associated genes.

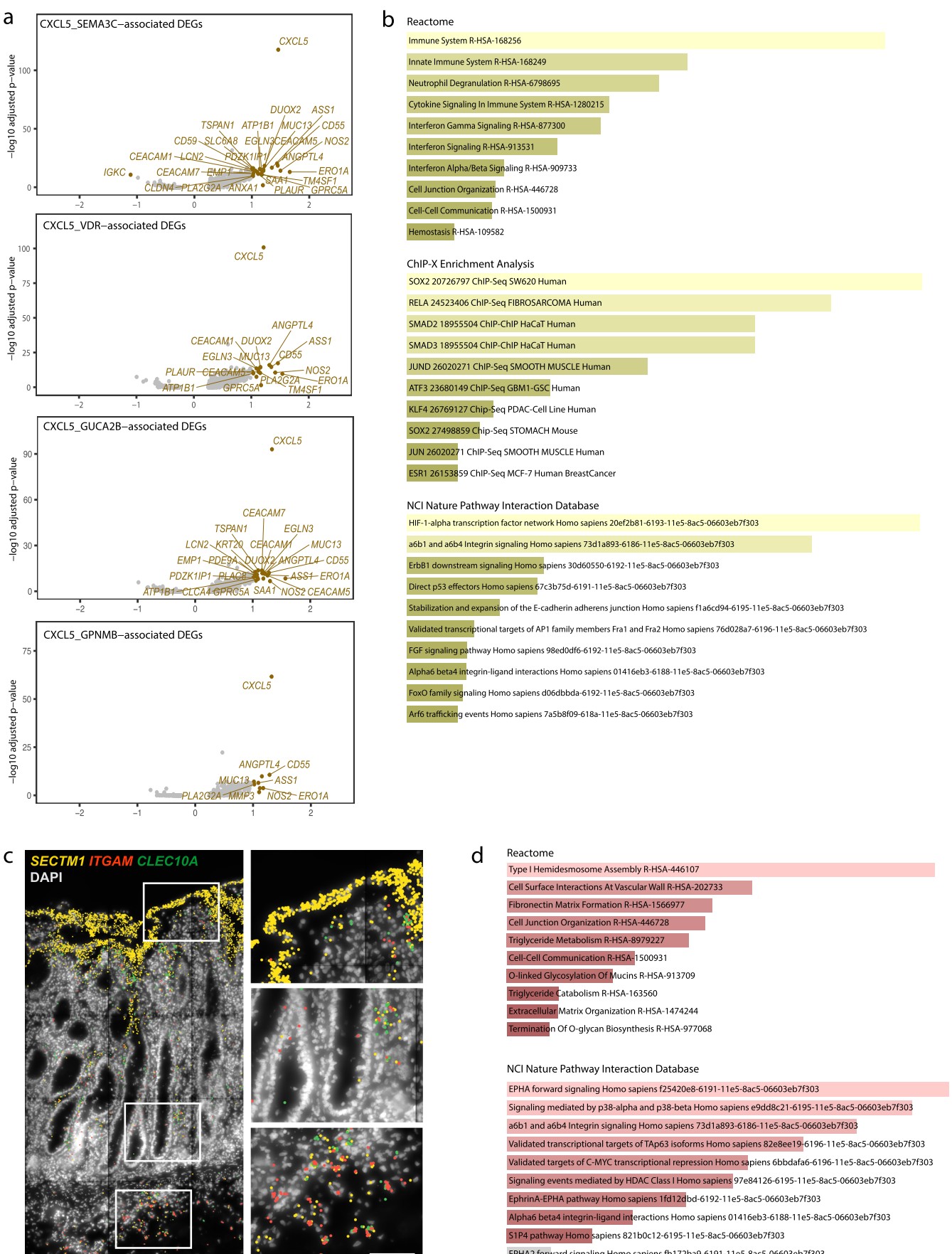

◀ **Figure EV4.   Differential co-occurrence analysis of the surfaceome reveals concerted tissue responses.**

(**A**) *CXCL5*-interactions associated DEGs. Non-parametric Wilcoxon rank sum test. (**B**) Reactome, ChiP-X Enrichment Analysis and NCI Nature pathway enrichment of shared *CXCL5*-associated DEGs. (**C**) Molecular Cartography image of *SECTM1, ITGAM* (neutrophils) and *CLEC10A* (DCs) expression. Scale bar, 20 µm. (**D**) Reactome and NCI Nature pathway enrichment of *SECTM1*-associated DEGs.

