## [Peer Review File · Molecular Systems Biology]

Identifying Spatial Co-occurrence in Healthy and Inflamed tissues (ISCHIA)

Atefeh Lafzi, Costanza Borrelli, Simona Baghai Sain, Karsten Bach, Jonas Kretz, Kristina Handler, Daniel Regan-Komito, Xenia Ficht, Andreas Frei, and Andreas Moor

DOI: [10.15252/msb.202312033](https://doi.org/10.15252/msb.202312033)

Corresponding author(s): *Andreas Moor (andreas.moor@bsse.ethz.ch)*

Review Timeline:

Transfer from Review Commons:	27th Sep 23
Editorial Decision:	3rd Nov 23
Revision Received:	28th Nov 23
Accepted:	8th Dec 23

Editor: Maria Polychronidou

Transaction Report: This manuscript was transferred to Molecular Systems Biology following peer review at Review Commons.

Review #1

1. Evidence, reproducibility and clarity:

Evidence, reproducibility and clarity (Required)

Summary:

In this manuscript Lafzi et al. present a novel computational framework (ISCHIA) for the analysis of spatial occurrence patterns, be it of cells or transcript species, found in spatial transcriptomics datasets. The authors also show its applications in finding differentially co-occurring ligand-receptor pairs, as well as inter-species analysis to find conserved cell signalling pathways. ISCHIA consists of a well-documented R package and utilizes empirical probabilistic estimations of non-random co-occurrence, as used in the field of ecology, which to my knowledge is novel in the field. The authors also validate their predictions using an orthogonal technology (in situ hybridization-based spatial transcriptomics), which is a nice addition to the computational work presented in the manuscript.

Major:

- When determining the composition classes, the authors discard 4 out of 8 clusters of composition classes, partly due to being highly patient-specific. It's unclear how sensitive ISCHIA is for batch effects which might affect the measured cellular fractions. Given that the presence of batch effects is highly likely with ST methods, due to the sample processing procedures, it would help the reader/potential user to estimate the impact these could have on the resulting output. It would also be useful to plot a version of the UMAP with sample labels as to see if the remaining clusters are properly mixed (at least between replicates of the same condition). Additionally, it would help to illustrate that the biological findings reported in the manuscript are supported across more than 1 biological replicate.
- In the LR analysis, the authors state that ISCHIA's predictions are agnostic to gene expression levels, as the authors model expression as a Boolean (gene count threshold > 0). Wouldn't low expression levels result in increased drop-out due to imperfect sensitivity? This would likely inflate false negative predictions at low expression levels.
- The authors show the enrichment of particular pathways/genesets in differential gene expression comparing interacting vs noninteracting spots (through LR expression) within the same CC. It is however unclear if this enrichment stems from a random sampling of the CC (with possible confounding factors such as batch effect, QC

metrics, which might also have a spatial component such as localized tissue degradation) or from the actual interaction. Adding a measure of uncertainty, such as by permuting over interaction-labels to generate a proper null distribution, would help the user to ascertain the robustness of the results. For clarity, it would also be good to add how this is exactly computed to the Methods section.

- It's unclear if the p-values in the manuscript are adjusted for multiple comparisons or not. Given the number of hypotheses being tested here, this is a crucial issue.
- The authors don't really mention any of the existing state-of-the-art methods (e.g. Squidpy, Spacemake, Giotto, ...). This doesn't necessitate a full benchmark, but at least the authors should then state qualitatively what the difference is between the chosen approach and already available packages, with their respective added advantages/disadvantages.

****Minor:****

- When a priori testing for LR interactions without restricting these interactions to predicted interactions, it would be informative to have an estimate of how many of the positively co-occurring interactions coincide with their predictions. As the authors state, it's hard to judge novel interactions without orthogonal validation, but a large overlap between predictions and the results presented here might instill confidence in the novel findings.
- Fig 4D: It's hard to judge very small p-values on this plot, might be better to plot $-\log_{10}(pval)$.
- The axes on some of the plots should be better defined in the figure legends (e.g. Fig 4D, 5C)

I'm not an expert in inflammation or IBD biology, so I will defer that to other reviewers more suited to comment on this.

2. Significance:

Significance (Required)

The proposed method provides a reasonable framework for studying co-occurrences of cell types and transcripts (particularly ligand-receptor pairs), which are currently questions of great interest to the community applying novel spatial transcriptomics technologies in many different domains of life sciences. The manuscript is very well written, and provides a clear and consistent logical flow. The manuscript can be easily read and understood both by specialized users as well as biologist/clinical end-users wanting to apply the proposed technique. The addition of experimental data using an orthogonal technology to validate computational predictions illustrates nicely the

power of the proposed approach.

Although the presented approach is methodologically rather simple (which is not necessarily a disadvantage), it is novel in the field as far as I know and a good implementation is likely to see great adoption by the field, especially if it's well documented, maintained and integrated into existing data processing workflows. The authors should however compare their approach fairly with the rest of the available packages in order to convince the reader.

Although the presented data seems convincing to me, the authors should take greater care of defining good practice statistical reporting of their findings. Even though these tools are often hypothesis-generating and predictions should always be experimentally validated, some end-users might interpret p-values literally. As such, proper multiple-testing correction and analysis of critical confounding factors should be carried out as to set an example.

I'm a computational biologist with expertise in method development (machine learning and statistical modelling) for spatial multi-omics assays. I'm not an expert in inflammation or IBD biology, so I will defer that to other reviewers more suited to comment on this.

3. How much time do you estimate the authors will need to complete the suggested revisions:

Estimated time to Complete Revisions (Required)

(Decision Recommendation)

Less than 1 month

Yes

Review #2

1. Evidence, reproducibility and clarity:

Evidence, reproducibility and clarity (Required)

The authors developed ISCHIA to study co-occurrence of cell types and transcript species. This work was further extended to study cell-cell interactions based on ligand-receptor co-expression. The observation by ISCHIA was further validated using hybridization based spatial transcriptomics approaches. ISCHIA was applied to study healthy and inflamed human colons.

****Referees cross-commenting****

As the reviewer #1 pointed, there is no description about existing methods. The reviewer #1 only asked stating qualitative differences.

If the manuscript is mainly for IBD and ISCHIA is the bioinformatics steps they followed, I would agree with the reviewer #1. However, the authors wanted to say that it is a new software. I still think that full benchmarking is needed in this circumstance.

2. Significance:

Significance (Required)

As the authors wanted to introduce ISCHIA as a new tool, discussion and comparison with the previous approaches are essential. The manuscript lacks discussion and the comparison with others. Co-localization has been discussed already in many articles including [PMID:325799]. It does not seem to require additional packages to study co-localization for cell type. There are many cell-cell interaction studies using ligand-receptor co-localization [ref; stLern, SpaGene, and many]. It is not well documented about the relationships with the previous works. Given the advances in algorithms for spatial transcriptomics, it is very uncertain that ISCHIA can provide additional knowledge or contribute to algorithmic development.

Previously, Visium data were generated by Elmentaite et al. (Nature 2021) against healthy and IBD samples. what are the new findings of the manuscript?

3. How much time do you estimate the authors will need to complete the suggested revisions:

Estimated time to Complete Revisions (Required)

(Decision Recommendation)

More than 6 months

No

Review #3

1. Evidence, reproducibility and clarity:

Evidence, reproducibility and clarity (Required)

****Summary****

The authors provide a framework to analyze spatial transcriptomics (ST) data in terms of spatial co-occurrence of cell types, and ligand-receptor pairs. The method was applied to an ulcerative colitis sequencing-based data set (10x Visium) and validated using a matched hybridization-based data set (Molecular Cartography).

****Major Comments****

The Visium data set consisted of a single slide with four samples. The authors should

clarify if the current implementation of their method is limited to a single Visium slide.

In Supplementary Table 1, I think it would be useful to include the minimum number of counts for the Ligand-Receptor genes. Given that the current threshold is 1, I think it warrants a discussion if the minimum number of counts has an effect on whether the ligand-receptor pair is significantly co-occurring (i.e. if ligand-receptor pairs with more counts are more likely to be significant).

Given the effect of outliers in the Pearson correlation and the nature of the expression values for Visium data, I think that the Spearman rank correlation is better suited to estimate the correlation between the expression values of the ligand-receptor pairs than the Pearson correlation (the default in R).

In the section titled "Differential co-occurrence identifies niche-specific response programs", it is unclear whether the spatial co-occurrence analysis was done within each CC.

****Minor Comments****

I found a few typos in the manuscript

In the Abstract, "tecniqee" instead of "techniques"

On page 10, under "Integration and annotation of scRNASeq data.", "W" instead of "We"

On page 11, there is an equation rendering error: $P(lt) = p_{lt}$

2. Significance:

Significance (Required)

The method proposed takes advantage of work done in ecology to leverage the spatial context of ST data. Furthermore, the methods proposed goes beyond describing spatial patterns present in the data, but allows for the comparison between two conditions of interest. The method proposed will be of interest to the growing number of researchers generating ST data.

My expertise is in statistical methods for single cell and spatial transcriptomics data.

Furthermore, I have extensive experience analyzing single cell and spatial transcriptomics data in the context of liver diseases.

3. How much time do you estimate the authors will need to complete the suggested revisions:

Estimated time to Complete Revisions (Required)

(Decision Recommendation)

Less than 1 month

Yes

Full Revision

Manuscript number: RC-2023-01881

Corresponding author(s): Andreas E Moor

[Please use this template only if the submitted manuscript should be considered by the affiliate journal as a full revision in response to the points raised by the reviewers.]

*If you wish to submit a preliminary revision with a revision plan, please use our "Revision Plan" template. **It is important to use the appropriate template to clearly inform the editors of your intentions.**]*

1. General Statements [optional]

Dear Editors,

Please find attached our revised manuscript entitled "Identifying Spatial Co-occurrence in Healthy and Inflamed tissues (ISCHIA)" which we are submitting for consideration in Molecular Systems Biology. In the present study, we propose the use of spatial co-occurrence analysis to gain deeper insights into cellular communities and the rules of their spatial associations.

In recent years, the advent of spatial transcriptomics (ST) has allowed profiling of the distribution of transcripts and cell types in the tissue. Sequencing-based ST techniques such as Visium (10X) allow unbiased capturing of RNA molecules at barcoded spots comprising 5-20 different cells. The coarse resolution of this technique is generally considered a disadvantage. However, we argue that the inherent proximity of transcriptomes captured on Visium spots can be leveraged to reconstruct cellular networks (CNs): communities of neighboring cells closely interacting to perform a coordinated function. CNs represent the basic functional unit of a tissue, hence understanding how they are altered by disease is important to understand pathology. Here, we present a computational framework that infers the architecture of CN by performing the following steps:

1. Deconvolution of spot transcriptomes: across all samples, every spot's transcriptome is deconvolved into the constituting cell types, both possible with and without single cell RNA sequencing reference.
2. Clustering of spots into composition classes (CCs): the deconvoluted matrix is clustered into groups of spots with similar cell type composition. Composition-based clustering of the tissue represents a major advantage of this method. Indeed, the downstream cell interaction predictions are conducted within individual CCs, filtering out transcriptomic differences arising from distinct cell type compositions.
3. Spatial co-occurrence analysis of cell types: identification of cell types whose observed co-occurrence in the tissue is significantly greater than what is expected by chance. Importantly, co-occurrence analysis is independent of cell type abundances, and only depends on spatial arrangement of cell types.
4. Spatial co-occurrence analysis of ligands and receptors: we apply co-occurrence analysis to ligand and receptor genes to prioritize interactions that spatially co-occur. We argue that due to the proximity of their expression, the encoded proteins are likely to interact via paracrine and juxtacrine signalling. As above, co-occurrence analysis of transcripts is independent of their expression levels, and only takes into account their distribution.
5. LR-pair associated transcriptional signatures: we compute differential gene expression between spots having and those lacking a given LR pair, deriving LR-associated gene signatures that grant insight into the function of CNs.

We apply this approach on Visium data of ulcerative colitis samples, and thereby uncover the emergence of an M cell-fibroblast cellular network in the inflamed human colon. We identify two putative

LR pairs that mediate M cell-fibroblasts interactions, and derive their associated gene signatures. To validate our findings at the single cell level, we generated matched 100-plex fluorescent in situ hybridization data (Molecular Cartography).

In conclusion, we show that co-occurrence analysis on spatial transcriptomics data can be used to chart the distribution and infer the interactions of cell types and transcripts in the tissue, revealing disease-specific cellular communities. It therefore represents a powerful tool for hypothesis-generation from spatial transcriptomics data. In this revised manuscript, we address reviewers' comments regarding multiple testing correction, batch effects and benchmarking against other methods.

We are grateful to the Referees for their detailed evaluation of our data and insightful remarks. We have now addressed all comments and amended the manuscript accordingly. Below we detail how we have addressed each point (responses are in *blue italic*). For your convenience we have also included the resulting changes in the manuscript in *blue*.

Reviewer #1 (Evidence, reproducibility and clarity (Required)):

Summary:

In this manuscript Lafzi et al. present a novel computational framework (ISCHIA) for the analysis of spatial occurrence patterns, be it of cells or transcript species, found in spatial transcriptomics datasets. The authors also show its applications in finding differentially co-occurring ligand-receptor pairs, as well as inter-species analysis to find conserved cell signalling pathways. ISCHIA consists of a well-documented R package and utilizes empirical probabilistic estimations of non-random co-occurrence, as used in the field of ecology, which to my knowledge is novel in the field. The authors also validate their predictions using an orthogonal technology (in situ hybridization-based spatial transcriptomics), which is a nice addition to the computational work presented in the manuscript.

Major:

When determining the composition classes, the authors discard 4 out of 8 clusters of composition classes, partly due to being highly patient-specific. It's unclear how sensitive ISCHIA is for batch effects which might affect the measured cellular fractions. Given that the presence of batch effects is highly likely with ST methods, due to the sample processing procedures, it would help the reader/potential user to estimate the impact these could have on the resulting output. It would also be useful to plot a version of the UMAP with sample labels as to see if the remaining clusters are properly mixed (at least between replicates of the same condition).

We thank the reviewer for the comment. We have included in Supplemental Figure S1c a UMAP colored by sample (patients), and a barplot in d) that illustrates how CC8 and CC4 are quite specific to patients 2 and 3. These graphs illustrate how the composition classes 1, 3, 5 and 6 are represented in all patients, albeit in different abundances. We would like to point out that the choice to exclude 4 out of 8 composition classes was not dictated by batch effects, but rather by the specific morphology of the samples available to us. Indeed, we excluded composition classes that were mapping to submucosal and muscular areas because only 2 out of 4 colon resections contained these anatomical compartments, and because we wanted to focus on cell type co-occurrences in the epithelial and subepithelial layers.

Additionally, in order to explain the sensitivity of ISCHIA for batch effects or sample-specific variation, we performed principal component (PC) analysis, and measured the standard deviation explained by each PC on the cell type deconvolution matrix (which is used to calculate composition clusters). Next, we tried to find an association of the covariate "batch" with the PCs that explain a high amount of variation in the data. In the deconvoluted matrix, while the first 4 PCs explain more than 80% of the variation in the data, we couldn't find association of the covariate "batch" to any of the first 4 PCs. We now include this analysis in the Result section:

Principal component analysis of the deconvolution matrix reveals no association of a particular sample with the first 4 principal components, which cumulatively explain 80% of the variance in the data (Supplemental Figure 2c, d). K-means clustering of the deconvolution matrix revealed 8 CCs of co-localizing cell types present in all samples (Fig 1c, d).

Supplemental Figure S2 | Composition-aware clustering of human colon Visium data. c, Percent of variance of the deconvolution matrix explained by the first 10 principal components. PC1-4 explain more than 80% of the variance. d, PC plot showing no association of any sample with a particular principal component.

While other methods assign a cell-type identity to spots based on the most abundant cell type detected by deconvolution algorithm, ISCHIA summarizes spot gene expression data in a presence-absence matrix. ISCHIA is therefore robust to variation in expression levels due to batch effects. This was also recently reported in the analytical tool Starfysh (He et al. 2022), which performs similar clustering of spots based on cell type composition. This is included in the Discussion:

While preserving the complexity of the cell type composition of the analyzed tissue, composition-based clustering of spots also confers robustness towards variations in expression levels due to batch effects. Indeed, other spatial analysis methods such as Starfysh⁶⁶ have found that finding inter-sample commonalities using composition-based clusters is easier compared to finding common transcriptome-based clusters between samples. Still, batch analysis and, if needed, correction of the ST data is recommended prior to analysis with ISCHIA.

Additionally, it would help to illustrate that the biological findings reported in the manuscript are supported across more than 1 biological replicate.

We agree with the reviewer that the sample size of Visium and Resolve datasets is limited, however greater than 1 (3 and 4 patients, respectively). ISCHIA is meant as a tool for hypothesis generation. Its findings require independent functional validation in model systems and bigger patient cohorts.

In the LR analysis, the authors state that ISCHIA's predictions are agnostic to gene expression levels, as the authors model expression as a Boolean (gene count threshold > 0). Wouldn't low expression levels result in increased drop-out due to imperfect sensitivity? This would likely inflate false negative predictions at low expression levels.

We agree with the reviewer that dropouts due to low capture rate of Visium will lead to false negative predictions. Indeed, we chose to set the threshold for gene count > 0 to account for the sparsity of captured transcripts. In single cell RNA sequencing analysis, gene expression is analyzed in cell clusters rather than in single cells. Similarly, ISCHIA calculates LR co-occurrence in composition classes, that is clusters of spots of similar cell composition. Aggregating spots in composition classes thus mitigates the effects of low capture rate and consequent false negative predictions. We now include a sentence explaining this concept in the Results section:

The count threshold is a user defined parameter that can be increased to restrict the co-occurrence analysis to highly expressed ligands and receptors. To account for the sparsity of ST data, ISCHIA calculates LR co-occurrence within composition classes, that is clusters of spots with similar cell mixtures. Aggregating spots in composition classes thus mitigates the effects of low transcript capture rate and consequent false negative predictions.

The authors show the enrichment of particular pathways/genesets in differential gene expression comparing interacting vs noninteracting spots (through LR expression) within the same CC. It is however unclear if this enrichment stems from a random sampling of the CC (with possible confounding factors such as batch effect, QC metrics, which might also have a spatial component such as localized tissue degradation) or from the actual interaction. Adding a measure of uncertainty, such as by permuting over interaction-labels to generate a proper null distribution, would help the user to ascertain the robustness of the results. For clarity, it would also be good to add how this is exactly computed to the Methods section.

We thank the reviewer for this remark and now perform a gene expression noise estimation. As suggested by the reviewer, we employed a permutation-based approach to assess the significance of differentially expressed genes (DEGs) identified by comparing spots that are double positive for expression of a ligand-receptor pair vs spots that are not expressing any of the genes in a specific CC. To do so, we performed 1000 random sampling of spots into two groups, and calculated DEGs between these groups, consequently generating a null distribution of DEGs. We next calculated Monte Carlo p-values for the LR-associated DEGs, comparing the initially computed p-values DEG with the null distribution, and adjusting them for multiple testing using FDR. Significant adjusted p-values were indicative of genes whose differential expression was robust and unlikely to result from random spot sampling.

From the Method section:

For the calculation of L-R-associated DEGs, ISCHIA computes differential gene expression between spots that are double positive or double negative for a given L-R pair. The significant DEGs are then used for pathway enrichment with any tool of choice, such as EnrichR (<https://bio.tools/enrichr>). We employed a permutation-based approach to assess the significance of the obtained DEGs. Specifically, we generated a null distribution of DEGs (noise estimation) by 1000-fold random sampling of spots into two groups, and calculating DEGs between these groups. Next FDR-adjusted Monte Carlo p-values were calculated for each LR-associated DEG, comparing the initially computed p-values DEG from with the null distribution, and subsequently adjusting for multiple testing. DEGs with Monte Carlo FDRs < 0.05 are likely specific to the presence of a given LR and unlikely to result from random spot sampling.

It's unclear if the p-values in the manuscript are adjusted for multiple comparisons or not. Given the number of hypotheses being tested here, this is a crucial issue.

We agree with the reviewer that this is a crucial issue. In the ecology papers we consulted and in the original co-occur R package (Griffith et al. 2016), multiple testing correction was not applied when computing co-occurrence of species. We believe however, that it is necessary to correct p values when computing co-occurrence of ligands and receptor pairs, and now implement FDR correction in the ISCHIA pipeline. We have amended all the relative figures and tables.

The authors don't really mention any of the existing state-of-the-art methods (e.g. Squidpy, Spacemake, Giotto, ...). This doesn't necessitate a full benchmark, but at least the authors should then state qualitatively what the difference is between the chosen approach and already available packages, with their respective added advantages/disadvantages.

We thank the reviewer for the remark and we have compared the analysis performed by ISCHIA with other state-of-the-art tools. The main difference between ISCHIA with respect to other methods such as Spacemake (Sztanka-Toth et al. 2022), Squidpy (Palla et al. 2022) and Giotto (Del Rossi et al. 2022), is that ISCHIA computes cell-type and LR co-occurrence within individual spots, not between neighboring

spots. We include here an extensive comparison with other methods for this reviewer, and now discuss the differences between ISCHIA and other tools in the manuscript.

For neighborhood analysis, Spacemake and Squidpy use spatial coordinates of spots to identify neighbors among them (neighborhood sets are defined as a fixed number of adjacent spots in a square or hexagonal grid). Squidpy computes co-occurrence of clusters in spatial dimensions, however it uses the coordinates of spots and clusters to calculate co-occurrence of entire clusters of spots, not of cell-types within spots. This approach ignores the missing data between spots, as well as the multicellular nature of each spot. Similarly to Squidpy, Giotto assigns a score of a cell type to each spot upon deconvolution, to further identify the spatial patterns of the major cell taxonomies across all the spots on the tissue. For image-based spatial technologies with single cell resolution, Giotto creates a neighborhood graph of the single-cells to study gene expression patterns. This is similar to the approach we used to validate our predictions from Visium data in the Resolve dataset.

Tangram (Biancalani et al. 2021) is a deep learning approach to harmonize sc/snRNA-seq data with *in situ*, histological, and anatomical data, toward a high-resolution, integrated atlas. Tangram focuses on learning spatial gene-expression maps transcriptome-wide at single-cell resolution, and relating those to histological and anatomical information from the same specimens. However, it does not address cell-cell and ligand receptor interactions, nor co-occurrences from spatial data. Therefore, Tangram can be used to improve the deconvolution step of ISCHIA, to improve the definition of cellular composition in multicellular spatial spots.

Starfysh (He et al. 2022) is a computational toolbox for joint modeling of ST and histology data, dissection of refined cell states, and systematic integration of multiple ST datasets from complex tissues. It uses an auxiliary deep generative model that incorporates archetypal analysis and any known cell state markers to avoid the need for a single-cell-resolution reference. Starfysh also clusters spots based on cell type composition, and terms group of spots with similar composition “spatial hubs”. They use spatial hubs to integrate multiple samples, and to uncover regions with varying composition of cell states. As we propose in ISCHIA, the Starfysh authors also suggest that finding inter-sample commonalities using spatial hubs is easier compared to finding common clusters between samples. Starfysh addresses the co-localization of cell states by calculating the spatial correlation index (SCI) within a certain hub and penalizing the calculated correlation with a weight matrix τ in a way that : $\tau_{(i,j)}=1$ if the coordinate distance of spot i and spot j was less than $\sqrt{3}$ else $\tau_{(i,j)}=0$. While this approach provides a measure of cell state co-localization across a spatial hub, it looks at the problem from an inter-spot perspective, similar to Squidpy and Giotto. Again, this is different from ISCHIA that calculates co-occurrence within spots.

In conclusion, none of the current approaches focuses on addressing the co-occurrence of cell types and molecules within individual Visium spots. As the analysis is fundamentally different, we did not perform a full quantitative benchmark. We agree, however, that these differences need to be addressed in the manuscripts. To illustrate the different results obtained, we ran ISCHIA on a Visium slide of a coronal section of the mouse brain, which was also analyzed using Squidpy (https://squidpy.readthedocs.io/en/stable/notebooks/tutorials/tutorial_visium_hne.html) and Giotto (https://rubd.github.io/Giotto_site/articles/mouse_visium_brain_201226.html#part-9-spatial-network). We clustered the spots based on cell composition and then ran celltype co-occurrence analysis within each composition class (Supplementary Figure 1).

From the Results section:

State-of-the-art analysis tools for Visium data often treat every spot as a single datapoint, and compute co-localization, network or cell-cell interactions analysis between neighboring spots (inter-spot analysis). We hypothesized that CNs would be best reconstructed within individual spots (intra-spot analysis), as their mixed transcriptome contains information about locally occurring cell types, expressed ligands and receptors, and activated signaling pathways. As inferring CNs in each individual spot separately would be noisy, sparse, computationally intensive, and would lack statistical power, ISCHIA first divides the tissue into clusters of spots with similar cellular composition - termed composition classes (CCs) (Fig 1a). CCs are thus groups of spots containing similar mixtures of cells, or cellular communities, e.g, all spots capturing colonic crypts. To achieve the division of the tissue into CCs, spot transcriptomes are deconvoluted, yielding a cell type composition matrix (spot \times contribution of each cell type),

Full Revision

which is then subjected to dimensionality reduction and k-means clustering. ISCHIA allows for both reference-based deconvolution, with tools such as SPOTlight² or RCTD³, and reference-free deconvolution². Upon deconvolution, ISCHIA summarizes spot gene expression data in a cell type presence-absence matrix, where each listed cell type is associated with a probability to be present in a given spot ($p > 0.1$). Each spot is thus represented as a mixture of cell types, and similar mixtures are then clustered together in CCs. We applied ISCHIA on a publicly available Visium slide of a coronal section of the mouse brain (10x Genomics), using as a reference for deconvolution a scRNA-seq dataset of ~14,000 adult mouse cortical cells from the Allen Institute¹⁰. Composition-based clustering of the spots yielded 5 CCs, which broadly reflect the annotated anatomical regions (**Supplemental Figure 1a**). ISCHIA then computes cell type co-occurrence for every CC separately, identifying spatial association of cells in close proximity (**Supplemental Figure 1b**). Intra-spot analysis reconstructs cellular networks with cell types as nodes, and is distinct from inter-spot networks analysis employed by other tools on this sample, in which spots are used as nodes^{11,12}.

a

b

Supplemental Figure S1 | Composition-aware clustering and cell type co-occurrence in mouse brain Visium data. **a**, A Visium sample of a mouse brain coronal section (10x Genomics) is deconvoluted using a scRNASeq reference¹⁰ and then subjected to composition-based clustering, yielding 5 composition classes. **b**, Diagonal matrix plot depicting cell type co-occurrences in every CC. Co-occurrence is positive when observed more frequently than expected ($P < 0.05$), random when there is no significant difference, negative when observed less than expected ($P < 0.05$).

We also discuss differences between ISCHIA and other tools in the Discussion:

ISCHIA differs from other analysis tools for Visium data in that it predicts CNs within spots and not across spots. Indeed, spot data from sequencing-based ST methods such as Visium, simultaneously captures information about 1) cell types, 2) expressed LR genes, and 3) associated transcriptional responses at multiple spatially restricted locations. As proximity is a prerequisite for juxtacrine and paracrine cell-cell communication, which in turn constitutes the basis for the coordinated function of CNs, we hypothesized that CNs would best be reconstructed within individual spots, rather than across neighboring spots. To increase robustness, spots are grouped in clusters of similar cellular composition, termed composition classes. Composition-based clustering of the tissue represents a major advantage of this method, and distinguished it from other methods, such as Squidpy¹¹ or Giotto¹², that assign an identity to each spot based on marker gene expression or on the most abundant cell type. While preserving the complexity of the cell type composition of the analyzed tissue, composition-based clustering of spots also confers robustness towards variations in expression levels due to batch effects. Indeed, other spatial analysis methods such as Starfish¹³ have found that finding inter-sample commonalities using composition-based clusters is easier compared to finding common transcriptome-based clusters between samples. Still, batch analysis and, if needed, correction of the ST data is recommended prior to analysis with ISCHIA. Composition-based clustering of spots allows to restrict downstream analysis to similar mixtures of cells, filtering out transcriptome heterogeneity arising from distinct cellular compositions, which might act as a confounder variable when performing differential gene expression or cell-cell interaction predictions.

To reconstruct CNs, ISCHIA performs co-occurrence analysis of cell types within CCs. Other tools build a neighborhood graph using spatial coordinates of spots and a fixed number of adjacent spots^{11,12,16}, and therefore ignoring the missing data between spots as well as the multicellular nature of each spot, ISCHIA leverages the inherent proximity of mixed transcriptomes within individual spots to infer cellular neighborhoods. Hence, the cell types within the spots, rather than the spots themselves, are the nodes of the CN. This approach allows for reconstruction of much smaller CNs, operating in close spatial proximity, a prerequisite for juxtacrine and paracrine signaling between cells. ISCHIA further predicts LR interaction as edges connecting cell types within spots, not across multiple spots. Finally, by integrating co-occurrence of cell types, co-occurrence of LR pairs, and associated gene signatures, ISCHIA infers CN function.

Minor:

When a priori testing for LR interactions without restricting these interactions to predicted interactions, it would be informative to have an estimate of how many of the positively co-occurring interactions coincide with their predictions. As the authors state, it's hard to judge novel interactions without orthogonal validation, but a large overlap between predictions and the results presented here might instill confidence in the novel findings.

We thank the reviewer for the comment and now label in green, in Fig 4a, the positively co-occurring interactions that are also predicted by Omnipath, NicheNet or CellTalkDB.

Fig 4D: It's hard to judge very small p-values on this plot, might be better to plot $-\log_{10}(pval)$.

We have now changed this plot to display the differential co-occurrence score, calculated as $FDR_{inflamed} - FDR_{non-inflamed}$.

The axes on some of the plots should be better defined in the figure legends (e.g. Fig 4D, 5C)

We have included better axis descriptions.

Full Revision

I'm not an expert in inflammation or IBD biology, so I will defer that to other reviewers more suited to comment on this.

Reviewer #1 (Significance (Required)):

The proposed method provides a reasonable framework for studying co-occurrences of cell types and transcripts (particularly ligand-receptor pairs), which are currently questions of great interest to the community applying novel spatial transcriptomics technologies in many different domains of life sciences. The manuscript is very well written, and provides a clear and consistent logical flow. The manuscript can be easily read and understood both by specialized users as well as biologist/clinical end-users wanting to apply the proposed technique. The addition of experimental data using an orthogonal technology to validate computational predictions illustrates nicely the power of the proposed approach.

Although the presented approach is methodologically rather simple (which is not necessarily a disadvantage), it is novel in the field as far as I know and a good implementation is likely to see great adoption by the field, especially if it's well documented, maintained and integrated into existing data processing workflows. The authors should however compare their approach fairly with the rest of the available packages in order to convince the reader.

Although the presented data seems convincing to me, the authors should take greater care of defining good practice statistical reporting of their findings. Even though these tools are often hypothesis-generating and predictions should always be experimentally validated, some end-users might interpret p-values literally. As such, proper multiple-testing correction and analysis of critical confounding factors should be carried out as to set an example.

I'm a computational biologist with expertise in method development (machine learning and statistical modelling) for spatial multi-omics assays. I'm not an expert in inflammation or IBD biology, so I will defer that to other reviewers more suited to comment on this.

Reviewer #2 (Evidence, reproducibility and clarity (Required)):

The authors developed ISCHIA to study co-occurrence of cell types and transcript species. This work was further extended to study cell-cell interactions based on ligand-receptor co-expression. The observation by ISCHIA was further validated using hybridization based spatial transcriptomics approaches. ISCHIA was applied to study healthy and inflamed human colons.

****Referees cross-commenting****

As the reviewer #1 pointed, there is no description about existing methods. The reviewer #1 only asked stating qualitative differences.

If the manuscript is mainly for IBD and ISCHIA is the bioinformatics steps they followed, I would agree with the reviewer #1. However, the authors wanted to say that it is a new software. I still think that full benchmarking is needed in this circumstance.

We thank the reviewer for the remark and we have compared the analysis performed by ISCHIA with other state-of-the-art tools. The main difference between ISCHIA with respect to other methods such as Spacemake (Sztanka-Toth et al. 2022), Squidpy (Palla et al. 2022) and Giotto (Del Rossi et al. 2022), is that ISCHIA computes cell-type and LR co-occurrence within individual spots, not between neighboring

spots. We include here an extensive comparison with other methods for this reviewer, and now discuss the differences between ISCHIA and other tools in the manuscript.

For neighborhood analysis, Spacemake and Squidpy use spatial coordinates of spots to identify neighbors among them (neighborhood sets are defined as a fixed number of adjacent spots in a square or hexagonal grid). Squidpy computes co-occurrence of clusters in spatial dimensions, however it uses the coordinates of spots and clusters to calculate co-occurrence of entire clusters of spots, not of cell-types within spots. This approach ignores the missing data between spots, as well as the multicellular nature of each spot. Similarly to Squidpy, Giotto assigns a score of a cell type to each spot upon deconvolution, to further identify the spatial patterns of the major cell taxonomies across all the spots on the tissue. For image-based spatial technologies with single cell resolution, Giotto creates a neighborhood graph of the single-cells to study gene expression patterns. This is similar to the approach we used to validate our predictions from Visium data in the Resolve dataset.

Tangram (Biancalani et al. 2021) is a deep learning approach to harmonize sc/snRNA-seq data with *in situ*, histological, and anatomical data, toward a high-resolution, integrated atlas. Tangram focuses on learning spatial gene-expression maps transcriptome-wide at single-cell resolution, and relating those to histological and anatomical information from the same specimens. However, it does not address cell-cell and ligand receptor interactions, nor co-occurrences from spatial data. Therefore, Tangram can be used to improve the deconvolution step of ISCHIA, to improve the definition of cellular composition in multicellular spatial spots.

Starfysh (He et al. 2022) is a computational toolbox for joint modeling of ST and histology data, dissection of refined cell states, and systematic integration of multiple ST datasets from complex tissues. It uses an auxiliary deep generative model that incorporates archetypal analysis and any known cell state markers to avoid the need for a single-cell-resolution reference. Starfysh also clusters spots based on cell type composition, and terms group of spots with similar composition "spatial hubs". They use spatial hubs to integrate multiple samples, and to uncover regions with varying composition of cell states. As we propose in ISCHIA, the Starfysh authors also suggest that finding inter-sample commonalities using spatial hubs is easier compared to finding common clusters between samples. Starfysh addresses the co-localization of cell states by calculating the spatial correlation index (SCI) within a certain hub and penalizing the calculated correlation with a weight matrix τ in a way that : $\tau_{(i,j)}=1$ if the coordinate distance of spot i and spot j was less than $\sqrt{3}$ else $\tau_{(i,j)}=0$. While this approach provides a measure of cell state co-localization across a spatial hub, it looks at the problem from an inter-spot perspective, similar to Squidpy and Giotto. Again, this is different from ISCHIA that calculates co-occurrence within spots.

In conclusion, none of the current approaches focuses on addressing the co-occurrence of cell types and molecules within individual Visium spots. As the analysis is fundamentally different, we did not perform a full quantitative benchmark. We agree, however, that these differences need to be addressed in the manuscripts. To illustrate the different results obtained, we ran ISCHIA on a Visium slide of a coronal section of the mouse brain, which was also analyzed using Squidpy (https://squidpy.readthedocs.io/en/stable/notebooks/tutorials/tutorial_visium_hne.html) and Giotto (https://rubd.github.io/Giotto_site/articles/mouse_visium_brain_201226.html#part-9-spatial-network). We clustered the spots based on cell composition and then ran celltype co-occurrence analysis within each composition class (Supplementary Figure 1x).

From the Results section:

State-of-the-art analysis tools for Visium data often treat every spot as a single datapoint, and compute co-localization, network or cell-cell interactions analysis between neighboring spots (inter-spot analysis). We hypothesized that CNs would be best reconstructed within individual spots (intra-spot analysis), as their mixed transcriptome contains information about locally occurring cell types, expressed ligands and receptors, and activated signaling pathways. As inferring CNs in each individual spot separately would be noisy, sparse, computationally intensive, and would lack statistical power, ISCHIA first divides the tissue into clusters of spots with similar cellular composition - termed composition classes (CCs) (Fig 1a). CCs are thus groups of spots containing similar mixtures of cells, or cellular communities, e.g, all spots capturing colonic crypts. To achieve the division of the tissue into CCs, spot transcriptomes are deconvoluted, yielding a cell type composition matrix (spot \times contribution of each cell type),

Full Revision

which is then subjected to dimensionality reduction and k-means clustering. ISCHIA allows for both reference-based deconvolution, with tools such as SPOTlight² or RCTD³, and reference-free deconvolution². Upon deconvolution, ISCHIA summarizes spot gene expression data in a cell type presence-absence matrix, where each listed cell type is associated with a probability to be present in a given spot ($p > 0.1$). Each spot is thus represented as a mixture of cell types, and similar mixtures are then clustered together in CCs. We applied ISCHIA on a publicly available Visium slide of a coronal section of the mouse brain (10x Genomics), using as a reference for deconvolution a scRNA-seq dataset of ~14,000 adult mouse cortical cells from the Allen Institute¹⁰. Composition-based clustering of the spots yielded 5 CCs, which broadly reflect the annotated anatomical regions (Supplemental Fig 1). ISCHIA then computes cell type co-occurrence for every CC separately, identifying spatial association of cells in close proximity (Supplemental Fig xx). Intra-spot analysis reconstructs cellular networks with cell types as nodes, and is distinct from inter-spot networks analysis employed by other tools on this sample, in which spots are used as nodes^{11,12}.

a

b

Supplemental Figure S1 | Composition-aware clustering and cell type co-occurrence in mouse brain Visium data. **a**, A Visium sample of a mouse brain coronal section (10x Genomics) is deconvoluted using a scRNASeq reference¹⁰ and then subjected to composition-based clustering, yielding 5 composition classes. **b**, Diagonal matrix plot depicting cell type co-occurrences in every CC. Co-occurrence is positive when observed more frequently than expected ($P < 0.05$), random when there is no significant difference, negative when observed less than expected ($P < 0.05$).

We also discuss differences between ISCHIA and other tools in the Discussion:

ISCHIA differs from other analysis tools for Visium data in that it predicts CNs within spots and not across spots. Indeed, spot data from sequencing-based ST methods such as Visium, simultaneously captures information about 1) cell types, 2) expressed LR genes, and 3) associated transcriptional responses at multiple spatially restricted locations. As proximity is a prerequisite for juxtacrine and paracrine cell-cell communication, which in turn constitutes the basis for the coordinated function of CNs, we hypothesized that CNs would best be reconstructed within individual spots, rather than across neighboring spots. To increase robustness, spots are grouped in clusters of similar cellular composition, termed composition classes. Composition-based clustering of the tissue represents a major advantage of this method, and distinguished it from other methods, such as Squidpy¹¹ or Giotto¹², that assign an identity to each spot based on marker gene expression or on the most abundant cell type. While preserving the complexity of the cell type composition of the analyzed tissue, composition-based clustering of spots also confers robustness towards variations in expression levels due to batch effects. Indeed, other spatial analysis methods such as Starfish¹⁶ have found that finding inter-sample commonalities using composition-based clusters is easier compared to finding common transcriptome-based clusters between samples. Still, batch analysis and, if needed, correction of the ST data is recommended prior to analysis with ISCHIA. Composition-based clustering of spots allows to restrict downstream analysis to similar mixtures of cells, filtering out transcriptome heterogeneity arising from distinct cellular compositions, which might act as a confounder variable when performing differential gene expression or cell-cell interaction predictions.

To reconstruct CNs, ISCHIA performs co-occurrence analysis of cell types within CCs. Other tools build a neighborhood graph using spatial coordinates of spots and a fixed number of adjacent spots^{11,12,16}, and therefore ignoring the missing data between spots as well as the multicellular nature of each spot, ISCHIA leverages the inherent proximity of mixed transcriptomes within individual spots to infer cellular neighborhoods. Hence, the cell types within the spots, rather than the spots themselves, are the nodes of the CN. This approach allows for reconstruction of much smaller CNs, operating in close spatial proximity, a prerequisite for juxtacrine and paracrine signaling between cells. ISCHIA further predicts LR interaction as edges connecting cell types within spots, not across multiple spots. Finally, by integrating co-occurrence of cell types, co-occurrence of LR pairs, and associated gene signatures, ISCHIA infers CN function.

Reviewer #2 (Significance (Required)):

As the authors wanted to introduce ISCHIA as a new tool, discussion and comparison with the previous approaches are essential. The manuscript lacks discussion and the comparison with others. Co-localization has been discussed already in many articles including [PMID:325799]. It does not seem to require additional packages to study co-localization for cell type. There are many cell-cell interaction studies using ligand-receptor co-localization [ref; stLern, SpaGene, and many]. It is not well documented about the relationships with the previous works. Given the advances in algorithms for spatial transcriptomics, it is very uncertain that ISCHIA can provide additional knowledge or contribute to algorithmic development.

We agree with the reviews that there is an increasing body of work addressing co-localization of cell type and cell-cell interactions in spatial transcriptomic data. However, we assert that the introduction of our method holds substantial relevance and adds value to the field, notwithstanding its ostensible simplicity. Our method has been well-received within the scientific community, indicating its applicability and potential significance in deciphering complex cellular ecosystems. We believe our approach offers a distinct conceptual advantage by enabling the analysis of cellular communities within individual Visium spots, rather than solely between them, allowing for a more refined exploration of cellular interactions and co-localizations within specific spatial domains.

Previously, Visium data were generated by Elmentaite et al. (Nture 2021) against healthy and IBD samples. what are the new findings of the manuscript?

Visium data generated by Elmentaite et al is from pediatric Crohn's disease, not adult Ulcerative colitis. Spatial analysis of IBD samples has been performed by Nanostring and by CODEX (Garrido-Trigo et al. 2022; Mayer et al. 2023). Publication of our datasets (both Visium and Resolve) will increase the body of patient data available to the community, and should be considered positively.

Here, we use our dataset to demonstrate the ability of ISCHIA to reconstruct cellular networks within Visium spots, and identify a M-cell-fibroblast network in inflamed regions of UC patients. We further identify differentially co-occurring LR pairs in the inflammatory CC5 centered around EDN1, SEMA3C, and CXCL5. We further reveal inflammation-induced, protective responses from the colonic crypt involving the complement cascade and the immuno-modulator SECTM1. Finally, we apply co-occurrence analysis to an independent mouse Visium dataset and uncover differentially co-occurring LR pairs shared between the inflamed human and murine colon. ISCHIA is an hypothesis generating tool, and its findings should be extensively characterized and validated in larger cohorts.

Reviewer #3 (Evidence, reproducibility and clarity (Required)):

Summary

The authors provide a framework to analyze spatial transcriptomics (ST) data in terms of spatial co-occurrence of cell types, and ligand-receptor pairs. The method was applied to an ulcerative colitis sequencing-based data set (10x Visium) and validated using a matched hybridization-based data set (Molecular Cartography).

Major Comments

The Visium data set consisted of a single slide with four samples. The authors should clarify if the current implementation of their method is limited to a single Visium slide.

We thank the reviewer for the remark and now state in the text that ISCHIA can be applied to any Visium, Resolve or other spatial transcriptomic dataset. Visium samples originating from different slides and datasets can be fed into the ISCHIA pipeline, as it is quite robust to batch effects. Indeed, while other methods assign a cell-type identity to spots based on the most abundant cell type detected by deconvolution algorithm, ISCHIA summarizes spot gene expression data in a presence-absence matrix. ISCHIA is therefore robust to variation in expression levels due to batch effects. This was also recently reported in the analytical tool Starfysh (He et al. 2022), which performs similar clustering of spots based on cell type composition. This is included in the Discussion:

While preserving the complexity of the cell type composition of the analyzed tissue, composition-based clustering of spots also confers robustness towards variations in expression levels due to batch effects. Indeed, other spatial analysis methods such as Starfysh^{sc} have found that finding inter-sample commonalities using composition-based clusters is easier compared to finding common transcriptome-based clusters between samples. Still, batch analysis and, if needed, correction of the ST data is recommended prior to analysis with ISCHIA.

In Supplementary Table 1, I think it would be useful to include the minimum number of counts for the Ligand-Receptor genes. Given that the current threshold is 1, I think it warrants a discussion if the minimum number of counts has an effect on whether the ligand-receptor pair is significantly co-occurring (i.e. if ligand-receptor pairs with more counts are more likely to be significant).

We agree with the reviewer that increasing the threshold to >1 will reduce the number of significantly co-occurring LR pairs, but also increase false negative predictions. We chose to set the threshold for gene count > 0 to account for the sparsity of captured transcripts. Indeed, dropouts due to low capture rate of Visium will lead to false negative predictions. In single cell RNA sequencing analysis, gene

expression is analyzed in cell clusters rather than in single cells. Similarly, ISCHIA calculates LR co-occurrence in composition classes, that is clusters of spots of similar cell composition. Aggregating spots in composition classes thus mitigates the effects of low capture rate and consequent false negative predictions. We now include a sentence explaining this concept in the Results section:

The count threshold is a user defined parameter that can be increased to restrict the co-occurrence analysis to highly expressed ligands and receptors. To account for the sparsity of ST data, ISCHIA calculates LR co-occurrence within composition classes, that is clusters of spots with similar cell mixtures. Aggregating spots in composition classes thus mitigates the effects of low transcript capture rate and consequent false negative predictions.

Given the effect of outliers in the Pearson correlation and the nature of the expression values for Visium data, I think that the Spearman rank correlation is better suited to estimate the correlation between the expression values of the ligand-receptor pairs than the Pearson correlation (the default in R).

We thank the reviewer for the comment and now rank positively co-occurring LR based on Spearman correlation. See Fig 3a and Supplementary Table 1.

Figure 3 | Reconstructing cellular networks from cell type co-occurrence, LR co-occurrence and corresponding transcriptomic signatures. a, Top 20 positively co-occurring LR pairs in inflammatory CC5 (observed co-occurrence > expected co-occurrence, $FDR < 0.05$). Ranking based on Spearman correlation of expression of ligand and receptor genes within spots.

In the section titled "Differential co-occurrence identifies niche-specific response programs", it is unclear whether the spatial co-occurrence analysis was done within each CC.

We now specify that we computed co-occurrence analysis of ligands and receptor genes in all spots of our dataset across all CCs. We now include FDR corrected p-values in this analysis. Only after computing this broad co-occurrence analysis, we focus on differential co-occurrence comparing conditions or composition classes.

Minor Comments

I found a few typos in the manuscript

In the Abstract, "tecniquee" instead of "techniques"

On page 10, under "Integration and annotation of scRNASeq data.", "W" instead of "We"

On page 11, there is an equation rendering error: $P(t) = \$p_{lt}$

We thank the reviewer for these comments and have now corrected the typos!

Reviewer #3 (Significance (Required)):

The method proposed takes advantage of work done in ecology to leverage the spatial context of ST data. Furthermore, the methods proposed goes beyond describing spatial patterns present in the data, but allows

Full Revision

for the comparison between two conditions of interest. The method proposed will be of interest to the growing number of researchers generating ST data.

My expertise is in statistical methods for single cell and spatial transcriptomics data. Furthermore, I have extensive experience analyzing single cell and spatial transcriptomics data in the context of liver diseases.

3rd Nov 2023

Manuscript Number: MSB-2023-12033

Title: Identifying Spatial Co-occurrence in Healthy and InflAmed tissues (ISCHIA)

Dear Andreas,

Thank you again for submitting your revised study to Molecular Systems Biology along with the referee reports from Review Commons. We have now heard back from reviewer #1 who was asked to evaluate your revised study. As you will see below, the reviewer is satisfied with the performed revisions and supports publication in Molecular Systems Biology. Before we formally accept the study for publication, we would ask you to address the editorial issues listed below:

- Please include 5 keywords.
- Please make sure that no colored text remains in the final version.
- Please provide a .doc version of the manuscript text (including legends for main figures and EV figures) and individual production quality figure files for the main and EV Figures (one file per figure). The figure legends should be provided all together at the end of the main text, after the References.
- We have replaced Supplementary Information by the Expanded View (EV format). In this case, all additional figures can be provided as EV Figures. Please provide one file per EV Figure. Their legends should be included in the manuscript text. For detailed instructions regarding expanded view please refer to our Author Guidelines: .
- Table S1 should be provided and called out in the text as Dataset EV1. Please include the description of the EV Dataset in the dataset file itself, i.e. in a separate tab if it is provided as .xls or in a README.txt file in a .zip folders if the Dataset is provided as .csv.
- Please include a "Disclosure and Competing Interests Statement" in the main text.
- All Materials and Methods need to be described in the main text. We would encourage you to use 'Structured Methods', our new Materials and Methods format. According to this format, the Material and Methods section should include a Reagents and Tools Table (listing key reagents, experimental models, software and relevant equipment and including their sources and relevant identifiers) followed by a Methods and Protocols section in which we encourage the authors to describe their methods using a step-by-step protocol format with bullet points, to facilitate the adoption of the methodologies across labs. More information on how to adhere to this format as well as downloadable templates (.doc or .xls) for the Reagents and Tools Table can be found in our author guidelines: . An example of a Method paper with Structured Methods can be found here:
- Please include a Data availability section describing how the data, code etc. have been made available. This section needs to be formatted according to the example below:
The datasets and computer code produced in this study are available in the following databases:
 - Chip-Seq data: Gene Expression Omnibus GSE46748 (<https://www.ncbi.nlm.nih.gov/geo/query/acc.cgi?acc=GSE46748>)
 - Modeling computer scripts: GitHub (<https://github.com/SysBioChalmers/GECKO/releases/tag/v1.0>)
 - [data type]: [full name of the resource] [accession number/identifier] ([doi or URL or identifiers.org/DATABASE:ACCESSION])
- For data quantification: please specify the name of the statistical test used to generate error bars and P values, the number (n) of independent experiments (specify technical or biological replicates) underlying each data point and the test used to calculate p-values in each figure legend. The figure legends should contain a basic description of n, P and the test applied. Graphs must include a description of the bars and the error bars (s.d., s.e.m.).
- Please format the reference list according to the MSB style i.e. in alphabetical order (not numerical) and listing the first 10 authors followed by et al. The authors names should be listed with "last name" followed by "initial(s)" like in the example below: Grosjean H, Breton M, Sirand-Pugnet P, Tardy F, Thiaucourt F, Citti C, Barré A, Yoshizawa S, Fourmy D, de Crécy-Lagard V et al (2014) Predicting the minimal translation apparatus: lessons from the reductive evolution of mollicutes. PLoS Genet 10: e1004363
- Please provide a "standfirst text" summarizing the study in one or two sentences (approximately 250 characters), three to four "bullet points" highlighting the main findings and a "synopsis image" (550px width and max 400px height, jpeg format) to highlight the paper on our homepage.
- When you resubmit your manuscript, please download our CHECKLIST (<https://bit.ly/EMBOPressAuthorChecklist>) and include

the completed form in your submission.

Please note that the Author Checklist will be published alongside the paper as part of the transparent process (<https://www.embopress.org/page/journal/17444292/authorguide#transparentprocess>).

Please resubmit your revised manuscript online, with a covering letter listing amendments and responses to each point raised by the referees. Please resubmit the paper ****within one month**** and ideally as soon as possible. If we do not receive the revised manuscript within this time period, the file might be closed and any subsequent resubmission would be treated as a new manuscript. Please use the Manuscript Number (above) in all correspondence.

Click on the link below to submit your revised paper.

Link Not Available

Kind regards,

Maria

Maria Polychronidou, PhD
Senior Editor
Molecular Systems Biology

If you do choose to resubmit, please click on the link below to submit the revision online before 3rd Dec 2023.

Link Not Available

IMPORTANT: When you send your revision, we will require the following items:

1. the manuscript text in LaTeX, RTF or MS Word format
2. a letter with a detailed description of the changes made in response to the referees. Please specify clearly the exact places in the text (pages and paragraphs) where each change has been made in response to each specific comment given
3. three to four 'bullet points' highlighting the main findings of your study
4. a 'standfirst text' summarizing in two sentences the study (approx. 250 characters)
6. a "thumbnail image" (width=211 x height=157 pixels, jpeg format), which can be used as 'visual title' to highlight your paper on our homepage.
7. Please include an author contributions statement after the Acknowledgements section (see <https://www.nature.com/msb/authors/index.html#Submission>)
8. When assembling figures, please refer to our figure preparation guideline in order to ensure proper formatting and readability in print as well as on screen:
<https://bit.ly/EMBOPressFigurePreparationGuideline>
See also figure legend guidelines: <https://www.embopress.org/page/journal/17444292/authorguide#figureformat>

*** PLEASE NOTE *** As part of the EMBO Publications transparent editorial process initiative (see our Editorial at <https://www.nature.com/msb/journal/v6/n1/full/msb201072.html>), Molecular Systems Biology will publish online a Review Process File to accompany accepted manuscripts. When preparing your letter of response, please be aware that in the event of acceptance, your cover letter/point-by-point document will be included as part of this File, which will be available to the scientific community. More information about this initiative is available in our Instructions to Authors. If you have any questions about this initiative, please contact the editorial office (msb@embo.org).

Reviewer #1:

The authors have addressed all my outstanding concerns/remarks thoroughly and adequately. They present a compelling tool to analyse spatial transcriptomics data, that can be highly valuable to the community. The adoption of the tool will largely depend on active maintenance, documentation and ease-of-use of the software package. As the authors state, the proposed methodology is sufficiently different from other published tools that it will be interesting to see how effectively complementary these approaches are in the study of cell-cell interactions.

Rev_Com_number: RC-2023-01881

New_manu_number: MSB-2023-12033

Corr_author: Moor

Title: Identifying Spatial Co-occurrence in Healthy and InflAmed tissues (ISCHIA)

All editorial and formatting issues were resolved by the authors.

8th Dec 2023

Manuscript number: MSB-2023-12033R

Title: Identifying Spatial Co-occurrence in Healthy and InflAmed tissues (ISCHIA)

Dear Andreas,

Thank you again for sending us your revised manuscript. We are now satisfied with the modifications made and I am pleased to inform you that your paper has been accepted for publication.

Kind regards,

Maria

Maria Polychronidou, PhD
Senior Editor
Molecular Systems Biology

Rev_Com_number: RC-2023-01881

New_manu_number: MSB-2023-12033R

Corr_author: Moor

Title: Identifying Spatial Co-occurrence in Healthy and InflAmed tissues (ISCHIA)